# Zika virus pathogenesis in rhesus macaques is unaffected by pre-existing immunity to dengue virus

Petraleigh Pantoja[1,2,*], Erick X. Pérez-Guzmán[2,*], Idia V. Rodríguez[1], Laura J. White[3], Olga González[1], Crisanta Serrano[2], Luis Giavedoni[4], Vida Hodara[4], Lorna Cruz[1], Teresa Arana[2], Melween I. Martínez[1], Mariah A. Hassert[5], James D. Brien[5], Amelia K. Pinto[5], Aravinda de Silva[3] & Carlos A. Sariol[1,2,6]

Zika virus (ZIKV) is a re-emerging virus that has recently spread into dengue virus (DENV) endemic regions and cross-reactive antibodies (Abs) could potentially affect ZIKV pathogenesis. Using DENV-immune serum, it has been shown *in vitro* that antibody-dependent enhancement (ADE) of ZIKV infection can occur. Here we study the effects of pre-existing DENV immunity on ZIKV infection *in vivo*. We infect two cohorts of rhesus macaques with ZIKV; one cohort has been exposed to DENV 2.8 years earlier and a second control cohort is naïve to flaviviral infection. Our results, while confirming ADE *in vitro*, suggest that pre-existing DENV immunity does not result in more severe ZIKV disease. Rather our results show a reduction in the number of days of ZIKV viremia compared to naïve macaques and that the previous exposure to DENV may result in modulation of the immune response without resulting in enhancement of ZIKV pathogenesis.

[1] Unit of Comparative Medicine, Caribbean Primate Research Center, University of Puerto Rico-Medical Sciences Campus, San Juan, Puerto Rico 00952, USA. [2] Department of Microbiology and Medical Zoology, University of Puerto Rico-Medical Sciences Campus, San Juan, Puerto Rico 00936, USA. [3] Department of Microbiology and Immunology, University of North Carolina-Chapel Hill, North Carolina 27599, USA. [4] Texas Biomedical Research Institute, San Antonio, Texas 78227, USA. [5] Department of Molecular Microbiology and Immunology, Saint Louis University School of Medicine, Saint Louis, Missouri 63104, USA. [6] Department of Internal Medicine, University of Puerto Rico-Medical Sciences Campus, San Juan, Puerto Rico 00936, USA. * These authors contributed equally to this work. Correspondence and requests for materials should be addressed to C.A.S. (email: carlos.sariol1@upr.edu).

Zika virus (ZIKV), a member of the Flaviviridae family, is a re-emerging mosquito-borne virus. It was discovered in Uganda (1947) and has been reported in the Americas since 2014 with a major outbreak in Brazil starting in 2015 (refs 1,2). Since then, it has spread to at least 38 countries in South America, Central America and the Caribbean islands[1]. ZIKV has been associated with neurological sequelae Guillain–Barré syndrome (GBS) in adults and microcephaly in newborns. Because of the explosive outbreak in South America and fast spreading in other parts of the world, the World Health Organization (WHO) declared ZIKV a serious public health emergency in February 2016 (ref. 3). Of particular concern is that ZIKV infection virulently occurs in areas previously exposed to dengue virus (DENV). ZIKV and DENV belong to the same virus family, and are transmitted mainly by the same mosquito vector (Aedes aegypti). Both viruses are closely related, with a homology of 51–54% in the amino acid sequence of their envelope proteins[4]. DENV occurs in four serotypes in nature. While primary infection with one serotype confers life-long homotypic immunity to its own future infections, a heterotypic secondary infection promotes more severe clinical manifestations leading to DENV-hemorrhagic fever and DENV-shock syndrome (DHF/DSS)[5,6]. This has been explained by different mechanisms including the antibody-dependent enhancement (ADE) hypothesis[7–9]. This mechanism suggests that antibodies (Abs), generated in response to the primary infection with DENV, are not of sufficient concentration or avidity to neutralize a secondary infection with DENV of a different serotype, and may even facilitate its replication. ADE has been confirmed and demonstrated to occur in vitro, and has also been shown to drive higher loads of DENV in animal models[10–12]. Another mechanism proposed for the development of DHF/DSS is the production of cross-reactive T-cell clones during the primary infection[13,14]. Recently, different groups have shown that serum from DENV-immune subjects can induce ZIKV ADE in vitro[15–17]. Coincidentally, the only published study to date that provides evidence for ZIKV infection causing GBS used a cohort of 42 cases in French Polynesia of whom, strikingly, 95% had pre-existing DENV immunity[18]. The incidence of microcephaly and other fetus abnormalities have been reported with varied frequencies in two different countries with ZIKV outbreaks. The two published epidemiological studies addressing this situation are a prospective cohort study from Brazil[19] and a retrospective study in French Polynesia[20]. The Brazilian cohort showed that 29% of women with ZIKV infection at any time during pregnancy had abnormalities in prenatal ultrasonography, while the French Polynesia study suggested that only 1% of fetuses and infants from women who had ZIKV infection during the first trimester of pregnancy developed microcephaly. These and other observations have stimulated discussion and raised questions about the role of pre-existing flavivirus immunity in ZIKV pathogenesis especially during pregnancy[21,22].

The interval between primary and secondary DENV infection is considered to be critical in determining disease severity[7,23–25]. Heterotypic immunity can be protective to new serotypes for up to 6 months. Enhanced disease is, typically, observed when people are exposed to second infections 12 months or more after a primary infection. Here we study how waning DENV immunity (humoral or cellular) affects ZIKV disease. Using a cohort of macaques that were experimentally infected with DENV 2.8 years earlier, we show that DENV immunity does not enhance ZIKV pathogenesis in primates.

## Results

### DENV-immune serum induces ADE of ZIKV in vitro. Two groups with four macaques in each group were infected with

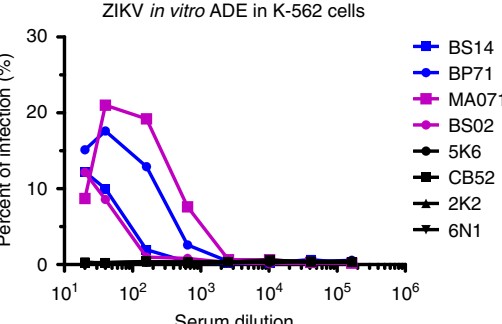

**Figure 1 | In vitro ADE of ZIKV infection by macaque DENV-immune antisera in K-562 cells.** Sera from four DENV-immune (two per serotype) and four naïve macaques were serially diluted and incubated with ZIKV strain H/PF/2013 at a multiplicity of infection (MOI) of 1. K-562 cells were added and incubated for 22 h, and the percentage of ZIKV-infected cells were determined by flow cytometry after intracellular staining with E-specific mAb 4G2 conjugated to Alexa-488. DENV-1 immune macaques are in blue, DENV-2 immune ones in magenta, and naïve ones in black. Data are representative of the average of triplicate samples per concentration in one experiment. Statistically significant differences among groups were calculated by two-way ANOVA using Bonferroni's multiple comparisons test.

DENV-1 or DENV-2. The animals were bled at 30 and 60 days and at 1, 1.5 and 2.5 years after DENV infection. To determine whether sera from DENV-immune macaques were able to enhance ZIKV infection in vitro, we performed an ADE assay on K562 cells. Sera collected from four DENV-immune macaques (two DENV-1 and two DENV-2 infected animals) collected 2.5 years after infection and sera from four flavivirus-naïve animals were used for the ADE assays. In the presence of naïve sera, ZIKV strain H/PF/2013 infected <1% of K562 cells (Fig. 1). Convalescent sera from DENV-immune macaques led to a substantial enhancement of ZIKV infection at low serum dilutions, ranging from 10 to 20% (Fig. 1). One DENV-2 pre-exposed animal (MA071, see below) showed the highest enhancement (>20%). These results confirm that previous exposure to DENV and the presence of DENV-immune serum induces ADE of ZIKV in vitro.

**Rhesus macaque cohorts.** In 2013, we infected rhesus macaques (*Macaca mulatta*) with $5 \times 10^6$ p.f.u. subcutaneously (s.c.) of DENV-1, Western Pacific 74, or DENV-2, New Guinea 44 (cohort 1 in Fig. 2). Both groups in this cohort were sequentially bled at 30 and 60 days and at 1, 1.5 and 2.5 years post-DENV infection. After that two macaques from each group were challenged with ZIKV for this proof-of-concept study 2.8 years after DENV infection. In addition, this work includes a second cohort (cohort 2) with four flavivirus-naïve macaques as control group. Prior to ZIKV infection all eight macaques were subjected to a forty-day quarantine period.

**Clinical status of macaques before and after ZIKV exposure.** ZIKV-infected macaques were regularly evaluated for evidence of disease or injury. No significant variations in weight (Fig. 3a) and external or rectal temperature (Fig. 3b,c) were detected. External temperature reading was above 38.8 °C only for one naïve animal (BS02) (Fig. 3b). Complete blood count (CBC) and chemistries were evaluated for all eight macaques on baseline and on days 7, 15 and 30 post infection (p.i.). White blood cell (WBC) counts dropped at 7 days post infection (d.p.i.) in both cohorts of macaques and returned to baseline levels by day 15 p.i. (Fig. 3d).

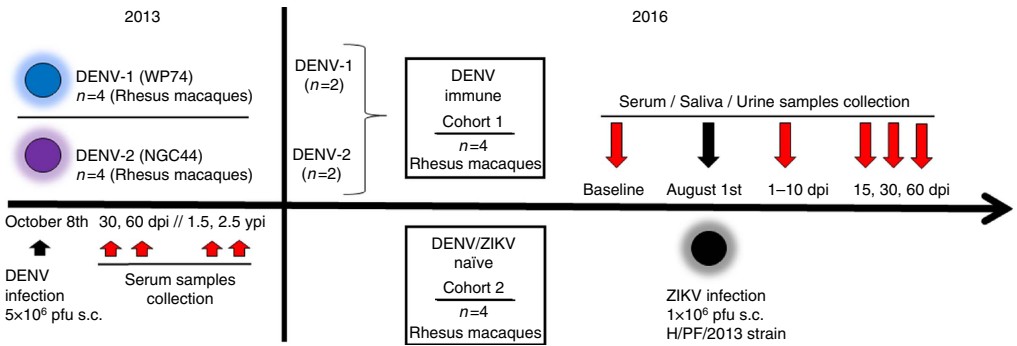

**Figure 2 | Experimental design of ZIKV infection in DENV-immune and naïve macaques.** Two cohorts of rhesus macaques (*Macaca mulatta*) were used. Cohort 1 was composed of four macaques previously infected with DENV ($5 \times 10^6$ p.f.u., s.c.) in 2013. Two macaques were from a group ($n = 4$) infected with DENV-1 Western Pacific 74 (WP74) strain (BS14 and BP71), and the other two macaques from a group ($n = 4$) infected with DENV-2 New Guinea 44 (NGC44) strain (MA071 and BS02). Cohort 1 in 2013 was sequentially bled at baseline, 30, 60 d.p.i., 1.5 and 2.5 years post DENV infection. Macaques in cohort 1 include one with high and one with low NAb titres against DENV-1 or DENV-2 (Supplementary Fig. 3). Cohort 2 contains four flavivirus-naïve macaques as control (5K6, CB52, 2K2 and 6N1). Macaques in both cohorts matched in age and sex. Cohorts were bled 30 days (Baseline) prior to infecting with ZIKV, and in August 1, 2016 they were infected with ZIKV ($1 \times 10^6$ p.f.u., s.c.) H/PF/2013 strain. Afterwards, all macaques were first bled from 1 to 10 d.p.i., and then on 15, 30 and 60 d.p.i. Furthermore, saliva and urine samples were also collected at the same timepoints. ZIKV infection was performed 2.8 years after DENV infection.

Absolute numbers of lymphocytes (LYM), neutrophils (NEU) and monocytes (MON) were also decreased at 7 d.p.i. both in naïve and pre-exposed macaques (Fig. 3e–g). However, compared to their baseline values, only the drop in the absolute NEU numbers at 7 d.p.i. in DENV-immune macaques was statistically significant ($P < 0.01$) (Fig. 3f). The percentage of WBC also changed after infection (Supplementary Fig. 1). The per cent of MON showed a significant increase in DENV-immune macaques by day 7 p.i. compared to their baseline values ($P < 0.05$) (Supplementary Fig. 1). Platelet counts remained constant in both cohorts (Fig. 3h). In all macaques, liver enzymes alanine aminotransferase (ALT) and aspartate aminotransferase (AST) levels were elevated at 7 d.p.i. when compared to baseline (Fig. 3i,j). Particularly, ALT showed a non-significant but strong trend towards higher values by day 7 p.i. in naïve macaques compared to both DENV-1 and 2-immune macaques. Values returned to baseline by day 15 (Fig. 3i,j). No apparent clinical signs were noticeable in seven of eight macaques. One of the four DENV-immune macaques (BS14) developed a nonpruritic skin rash (Supplementary Note 1, Supplementary Fig. 2).

**Immune profile of cohorts before ZIKV infection.** We measured the levels of binding and neutralizing Abs (Nabs) to DENV and ZIKV before challenging the animals with ZIKV. All DENV-immune macaques showed an increase of DENV IgG titres (Supplementary Fig. 3a). Reading of optical density (OD, 450 nm) that measures the level of IgG remained high up to 1 year after DENV infection, and declined slightly by 2.5 years (by the time the sample was collected prior to ZIKV infection) in all but two macaques infected with DENV-1 or DENV-2 (BP71 and MA071, respectively). All macaques were negative for DENV IgM (Supplementary Fig. 3b). DENV-naïve macaques were negative for both DENV IgM or IgG (Supplementary Fig. 3b,c) and for ZIKV-IgM (Supplementary Fig. 3f). Macaques in group 1 (DENV-1 immune) showed no reactivity against ZIKV-NS1 protein. From this group, two were selected for infection with ZIKV (BS14 and BP71). However, macaques pre-exposed to DENV-2 ($n = 4$), showed an increase of ZIKV-NS1 IgG at days 30 and 60 after DENV infection. Three out of four macaques in group 2 (DENV-2 immune) still had detectable cross-reactive IgG to ZIKV-NS1 protein at 1 and 2.5 years p.i. (Supplementary Fig. 3d). From group 2, one macaque with high cross-reactivity

(MA071) and one animal with undetectable levels of IgG to ZIKV-NS1 protein (BS02) at the time of the challenge, were selected for infection with ZIKV. As shown in Table 1, before ZIKV challenge, DENV-pre-exposed macaques had Abs that neutralized one or more DENV serotypes but not to ZIKV. In summary, DENV-infected animals had variable levels of IgG that cross-reacted with ZIKV antigens but the Nabs induced by DENV infections did not cross-neutralize ZIKV.

**Immune profile after ZIKV infection.** Thirty days after ZIKV infection, all exposed macaques showed a significant increase in the cross-reacting anti-DENV IgG value. The values of IgG anti-ZIKV NS1 protein were also enhanced in both cohorts (Supplementary Fig. 3e). All macaques developed IgM to ZIKV by 10 d.p.i. (Supplementary Fig. 3f). IgM titres peaked by day 15 and declined to basal levels by day 30 p.i. The naïve macaques challenged with ZIKV had high levels of Abs that neutralized ZIKV and also lower levels of DENV cross-neutralizing Abs that declined between day 30 and 60 after infection (Table 1). The DENV-immune macaques challenged with ZIKV developed high levels of ZIKV neutralizing antibodies as well as a boost in their DENV Nab titres (Table 1). In most of pre-immune macaques, the DENV and to a lesser extent the ZIKV neutralizing antibodies declined between days 30 and 60 p.i. (Table 1). The hierarchy of NAb titres observed 30 days after ZIKV infection was different in the two cohorts and also different for the two groups exposed to different DENV serotypes. For naïve macaques it was ZIKV > D2 > D1 > D3 > D4, for DENV-1 ZIKV > D1 > D2 > D4 > D3 and for DENV-2 immune macaques ZIKV > D2 > D3 > D1 > D4.

To further characterize the degree of Ab cross reactivity between DENV and ZIKV, we measured the binding of antibodies in the sera from the two cohorts before and after ZIKV infection to the whole ZIKA virus antigen using IgG ELISA. At baseline, all DENV-immune macaques showed cross-reactivity with titres up to a dilution of $1 \times 10 \log_3$ while naïve macaques were essentially negative (Fig. 4a). For samples collected 30 d.p.i., naïve macaques had ZIKV-specific IgG up to a dilution of $1 \times 10 \log_3$ and the DENV-immune cohort showed significantly higher values in the first two dilutions ($P < 0.05$) when compared to naïve macaques (Fig. 4b). DENV-immune macaques also showed a statistically significant increase in the OD values while measuring IgG endpoint

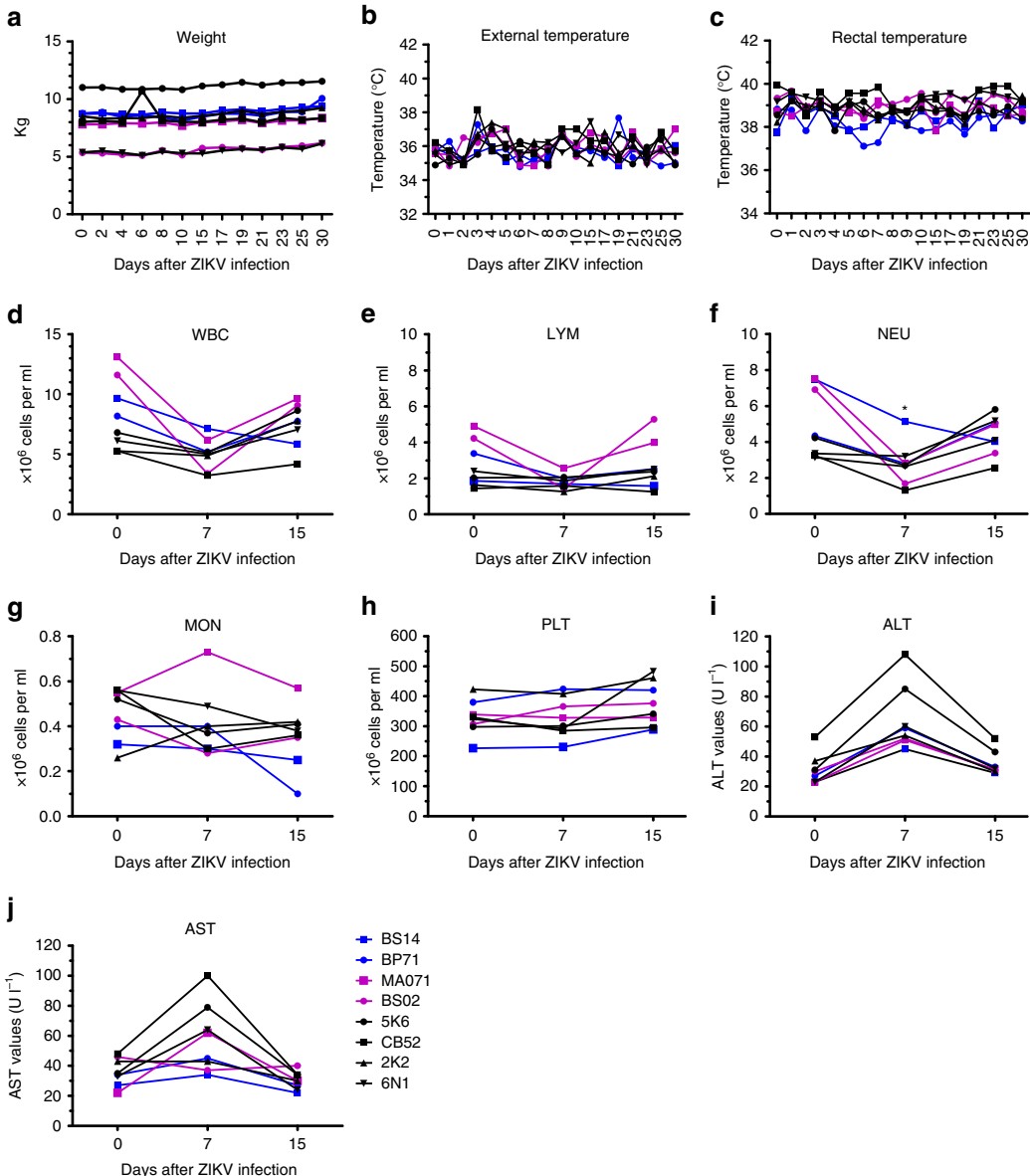

**Figure 3 | Vital signs and clinical status of macaques before and after ZIKV infection.** (a) Weight expressed in kilograms (kg). (b) Corporal temperature (in Celsius) measured using an infrared device. (c) Rectal temperature (in Celsius) was also measured. (d–g) Cell subsets: white blood cell (WBC), lymphocyte (LYM), neutrophil (NEU) and monocyte (MON) kinetics obtained from complete blood cell (CBC) counts ($10^6$ cells per ml) at baseline and on days 7 and 15 p.i. in absolute numbers. Comparison of absolute NEU numbers within cohorts at day 7 related to their own baseline values was performed using a two-tailed unpaired *t*-test with Sidak–Bonferroni correction ($P < 0.01$). (h–j) Levels of platelets ($10^6$ PLT per ml), Alanine Aminotransferase (ALT) and Aspartate Aminotransferase (AST) at baseline, 7 and 15 d.p.i. In all panels, the DENV-1 immune macaques are in blue, DENV-2 immune ones in magenta, and naïve ones in black.

titres for binding to ZIKV when compared to their own OD baseline values in the first two dilutions ($P < 0.05$ and $P < 0.01$). One of four macaques (MA071, DENV-2-exposed) showed the highest OD for serum dilution up to $1 \times 10$ $\log_5$. Remarkably, it is the same animal that showed cross reactivity with ZIKV-NS1 protein and the highest DENV NAb titres prior to the ZIKV infection. Altogether, these results confirm the contribution of pre-existing DENV-immunity to the expansion of cross-reacting ZIKV-IgG levels early after ZIKV infection in macaques.

**Viremia in serum**. To determine if pre-existing DENV immunity enhances ZIKV replication in primates, we measured ZIKV viremia using quantitative real-time reverse transcription PCR (qRT-PCR). Viral RNA (vRNA) detection was consistent in both

groups during the first 4 days after infection (Table 2, Fig. 5). On day 5 p.i., three of four naïve versus one of four DENV-immune macaques had detectable vRNA. On day 6 this ratio changed to two of four of naïve and three of four DENV-immune macaques with detectable viremia. DENV-immune macaques showed a trend with a shorter duration of viremia with only two out of four macaques having detectable levels of vRNA during the period of 7–15 d.p.i. In comparison, all four of naïve macaques had detectable vRNA levels in at least one day within the 7–15 d.p.i. By day 30 p.i., only two of the four DENV immune compared to all four naïve macaques had detectable ZIKV RNA. In general, DENV-immune macaques showed less viremia days compared to naïve macaques (25 versus 31 days, respectively). Of relevance to this study, the pre-challenge neutralization titres to DENV do not correlate with the peak or magnitude of ZIKV viremia (Table 1).

**Table 1 | Neutralizing titres against DENV and ZIKV before and after 30 and 60 days of ZIKV infection.**

| RM ID | DENV exposure history | PRNT/FRNT$_{60}$ titre 30 days before ZIKV infection | | | | | PRNT/FRNT$_{60}$ titres 30/60 days post ZIKV infection | | | | |
|---|---|---|---|---|---|---|---|---|---|---|---|
| | | ZIKV | DENV-1 | DENV-2 | DENV-3 | DENV-4 | ZIKV | DENV-1 | DENV-2 | DENV-3 | DENV-4 |
| *Cohort 1* | | | | | | | | | | | |
| BS14 | 1° DENV-1 | <1:20 | 1:80 | 1:20 | <1:20 | 1:80 | 1:2,560/1:640 | 1:640/1:160 | 1:320/1:320 | 1:160/1:80 | 1:160/1:80 |
| BP71 | 1° DENV-1 | <1:20 | 1:640 | 1:40 | 1:160 | 1:80 | 1:1,280/1:640 | 1:2,560/1:640 | 1:640/1:160 | 1:160/1:160 | 1:320/1:320 |
| MA071 | 1° DENV-2 | <1:20 | 1:40 | 1:2,560 | 1:160 | 1:80 | 1:5,120/1:2,560 | 1:160/1:160 | 1:2,560/1:2,560 | 1:320/1:320 | 1:80/1:640 |
| BS02 | 1° DENV-2 | <1:20 | <1:20 | 1:320 | <1:20 | 1:20 | 1:1,280/1:640 | 1:160/1:80 | 1:1,280/1:2,560 | 1:160/1:160 | 1:320/1:80 |
| *Cohort 2* | | | | | | | | | | | |
| 5K6 | Naïve | <1:20 | <1:20 | <1:20 | <1:20 | <1:20 | 1:1,280/1:1,280 | <1:20/<1:20 | 1:40/1:40 | <1:20/<1:20 | 1:160/1:40 |
| CB52 | Naïve | <1:20 | <1:20 | <1:20 | <1:20 | <1:20 | 1:1,280/ 1:1,280 | <1:20/<1:20 | <1:20/<1:20 | <1:20/<1:20 | 1:160/1:40 |
| 2K2 | Naïve | <1:20 | <1:20 | <1:20 | <1:20 | <1:20 | 1:2,560/ 1:2,560 | <1:20/<1:20 | 1:20/1:80 | <1:20/<1:20 | 1:40/1:40 |
| 6N1 | Naïve | <1:20 | <1:20 | <1:20 | <1:20 | <1:20 | 1:2,560/ 1:1,280 | 1:40/<1:20 | 1:160/1:40 | 1:80/<1:20 | 1:640/1:160 |

Neutralization of DENV (DENV-1 WP74, DENV-2 NGC44, DENV-3 Sleman 73 and DENV-4 Dominique) and ZIKV strain H/PF/2013 in Vero 81 cells in presence of the sera from DENV-immune or naïve macaques at baseline, 30 and 60 d.p.i. Data presented as the concentration that resulted in 60% reduction of plaque-forming units (p.f.u.) in a Focus/Plaque-Reduction Neutralization Test (FRNT$_{60}$ or PRNT$_{60}$) for DENV and ZIKV, respectively.

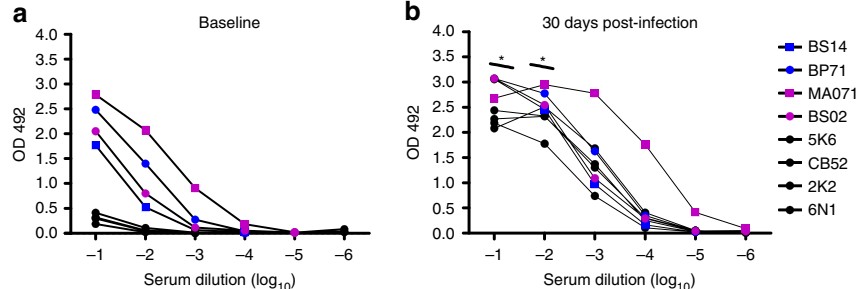

**Figure 4 | DENV-immune serum cross-reacts with ZIKV.** (**a**) Binding of baseline samples from the naïve (black circles) and the DENV-immune macaques (blue and magenta symbols for DENV-1 and DENV-2 immune macaques, respectively) assayed in six concentrations of serum and expressed as OD values at 492 nm. (**b**) Binding of samples collected 30 d.p.i. is represented as in **a**. In the first two-endpoints, serum from previous immunity to DENV resulted in significantly higher OD values ($P < 0.01$ and $P < 0.05$). Data are representative of duplicate samples per concentration in one experiment. Comparison of ZIKV IgG titres, expressed as OD values, within cohorts related to their own baseline values was performed using a two-tailed unpaired $t$-test with Sidak–Bonferroni correction (*$P < 0.05$).

**Table 2 | ZIKV viremia of naïve and DENV-immune rhesus macaques.**

| RM ID | DENV exposure history | Viremia (Log10 genome copies per ml) days post-ZIKV infection | | | | | | | | | | | | Total days |
|---|---|---|---|---|---|---|---|---|---|---|---|---|---|---|
| | | 1 | 2 | 3 | 4 | 5 | 6 | 7 | 8 | 9 | 10 | 15 | 30 | |
| *Cohort 1* | | | | | | | | | | | | | | |
| BS14 | 1° DENV-1 | 7.2 | 7.3 | 6.4 | 3.6 | ND | ND | ND | ND | ND | ND | ND | 2.9 | |
| BP71 | 1° DENV-1 | 6.0 | 7.1 | 6.5 | 4.0 | ND | 3.2 | 3.3 | ND | ND | ND | ND | ND | |
| MA071 | 1° DENV-2 | 6.7 | 7.6 | 7.0 | 5.5 | 3.3 | 3.0 | ND | 2.9 | ND | ND | 2.4 | 2.7 | |
| BS02 | 1° DENV-2 | 6.5 | 7.3 | 5.1 | 3.2 | ND | 3.2 | ND | ND | ND | ND | ND | ND | 25 |
| *Cohort 2* | | | | | | | | | | | | | | |
| 5K6 | Naïve | 6.5 | 7.0 | 5.8 | 3.8 | 3.5 | ND | ND | ND | ND | ND | 2.8 | 2.7 | |
| CB52 | Naïve | 6.4 | 7.3 | 7.2 | 5.7 | 3.7 | ND | 2.9 | ND | ND | ND | 2.8 | 2.7 | |
| 2K2 | Naïve | 6.8 | 7.8 | 6.4 | 4.2 | 4.0 | 3.7 | 4.1 | ND | ND | 3.0 | ND | 2.8 | 31 |
| 6N1 | Naïve | 6.9 | 7.3 | 6.1 | 3.4 | ND | 2.9 | ND | 2.9 | ND | ND | ND | 3.0 | |

ND, not detected.

**ZIKV in urine and saliva.** ZIKV vRNA was detected using qRT-PCR in the urine of macaques in both groups. However, the vRNA detection was transient throughout the period of 1–30 d.p.i. and considerably lower when compared to titres detected in the serum (Supplementary Table 1). Amplification of vRNA was observed for at least one or two days ($10^2$–$10^4$ genome copies per ml) in all naïve and DENV-immune macaques during 1–9 d.p.i. At day 15 p.i., one out of four pre-exposed macaques had detectable vRNA, and at day 17 one naïve and one pre-exposed macaques still had detectable vRNA. From day 19 to day

30 p.i. vRNA was not detectable in any animal. In addition to the samples that allowed clear detection of amplification of vRNA, there were several other samples that were highly suspicious for the presence of ZIKV. Although the vRNA amplification under the detection limit cannot be said as positive, we consider these samples as highly suspicious for containing ZIKV vRNA because they had positive amplification curves in terms of shapes and behaviour when compared to negative non-amplification curves (Supplementary Table 1). Within the limits of detection of our assays (up to 500 genome copies per ml), there was no amplification of vRNA in saliva.

**Immune cell subsets phenotyping.** To gain an understanding of pre-existing DENV immunity in macaques mounting a response against ZIKV, we analysed the cells involved in cellular immunity. Compared to the basal level, the frequency of activated B cells ($CD20^+/CD3^-/CD14^-/CD69^+$) was statistically higher in the naïve cohort at 24 and 48 h ($P < 0.05$) and 10 d.p.i. ($P < 0.01$) with a significant decrease by day 7 p.i. ($P < 0.01$) (Fig. 6a). DENV-immune macaques showed a shorter pattern of early activation. In this group, B cells were significantly activated after 24 h ($P < 0.05$) but not at 48 h followed by a decline on day 7 p.i. and a new peak frequency of activation on day 10 p.i. ($P < 0.05$). In the naïve cohort, the frequency of activation of these cells returned to baseline levels by day 15 p.i. while the pre-exposed cohort still had a positive but non-significant trend higher compared to their baseline levels (Fig. 6a).

While both cohorts showed activation of $CD4^+$ T cells with higher frequency of $CD3^+/CD4^+/CD69^+$ compared to the baseline levels, only naïve cohort showed statistically significant differences and longer periods of activation (Fig. 6b). In this cohort significant high frequency of activated cells were found at 24 and 48 h and on days 10 and 15 p.i. ($P < 0.05$, $P < 0.01$, $P < 0.01$ and $P < 0.05$, respectively) with a transient decrease by day 7 p.i. Compared to naïve macaques, DENV-immune cohort showed a limited pattern of activation with statistically significant high frequency of $CD3^+/CD4^+/CD69^+$ cells only at 24 and 48 h p.i. ($P < 0.05$) followed by a significant drop by day 7 p.i. ($P < 0.01$), which did not recover by day 10 p.i. (Fig. 6b). Because of that naïve macaques showed higher frequency of activated $CD4^+$ T cells at 48 h and on day 7 p.i. ($P < 0.05$ and $P < 0.01$, respectively), when compared to DENV-immune ones, in spite of the decrease by day 7 p.i. in both groups. Compared to baseline, naïve macaques showed a continued activation of $CD8^+$ T cells with significantly higher frequency of $CD3^+/CD8^+/CD69^+$ cells at 24 and 48 h and on days 7 and 10 p.i. ($P < 0.01$, $P < 0.01$, $P < 0.05$ and $P < 0.05$, respectively) (Fig. 6c). In contrast, although DENV-immune macaques showed a trend to increase the frequency of these cells, the differences compared to baseline values were not statistically significant at any measured time point (Fig. 6c). In both cohorts activated natural killer (NK) $CD16^+$ cells increased at 48 h after the infection, with a decrease in the frequency of activation by day 7 p.i., returning to baseline level by day 15 p.i. However, differences in the frequency of activation were not significant in any cohort (Supplementary Fig. 4). Nevertheless, the pattern of activation of NK $CD16^+$ $CD69^+$ we observed in our macaques is very similar to the pattern previously reported for rhesus macaques of Indian origin[26].

**T-cell immune response analysis.** To determine if a prior exposure to DENV impacted the ZIKV-specific functional response from the $CD8^+$ and $CD4^+$ T cells we measured their responses 30 d.p.i. of ZIKV in all eight macaques from both naïve and DENV-immune cohorts. To assess the stability of the

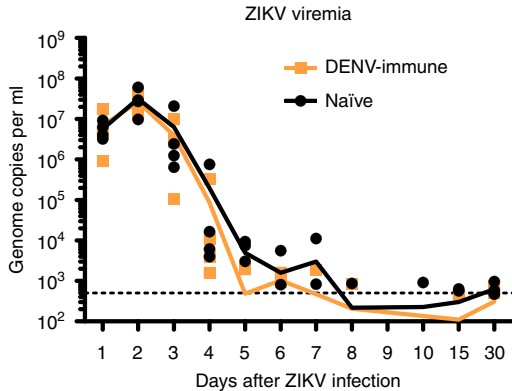

**Figure 5 | ZIKV RNA kinetics in serum.** ZIKV replication detected in serum in the first 10 days p.i. and on days 15 and 30 p.i. Genome copies per ml in blood serum samples from individual macaques are shown logarithmically. Orange and black lines represent mean values for the DENV-immune and naïve macaques, respectively. The limit of detection, 500 genome copies per ml, of our assay is indicated in black dotted line.

functional effector response we also looked at their effector responses 60 d.p.i. in three of the naïve ones where samples available (2K2, 5K6 and CB52) and all of the four DENV-immune (BP71, BS14, BS02 and MA071) macaques. Figure 7 shows the percentage of $CD8^+$ T cells from the PBMCs isolated on 30 d.p.i. that were capable of producing IFN-γ (Fig. 7a), TNF-α (Fig. 7b) and CD107a (Fig. 7c) in response to the listed stimuli. Figure 7d–f depicts the same effector function in the same macaques 60 d.p.i. of ZIKV allowing to assess the durability of such response. Unstimulated control levels for each animal carried out on days 30 and 60 d.p.i. are shown in Supplementary Table 2.

We first looked at the IFN-γ production from the $CD8^+$ T cells (Fig. 7a) 30 d.p.i. of ZIKV. We detected IFN-γ in the $CD8^+$ T cells from two of the DENV-immune macaques BP71 (0.56%) and MA071 (0.35%) in response to whole ZIKV stimulation (Fig. 7a). These macaques maintained this response until 60 d.p.i. (BP71, 0.42% and MA071, 0.24%) (Fig. 7d). We did not, however, detect a $CD8^+$ T-cell response to the DENV Envelope (E) peptide pool in any of the immune macaques suggesting that their response was not directed towards the DENV envelope. Two naïve (5K6, 0.32% and 6N1, 0.41%) and one pre-exposed (BS14, 0.62%) had detectable levels of IFN-γ following ZIKV-E peptide pool stimulation on day 30 p.i. However, by day 60, the peptide specific responses from 5K6 (0.06%) and BS14 (0%) were below the limit of detection. Three of the four naïve 2K2 (0.59%), 6N1 (0.71%), and CB52 (0.41%) and three of the four immune BP71 (0.53%), BS14 (0.46%), and MA071 (0.52%) macaques had detectable IFN-γ response to ZIKV infection on day 30; and on day 60 we were still able to detect a small response in one naïve 2K2 (0.31%) and two immune BP71 (0.34%) and MA071 (0.42%) macaques.

We focused next on TNF-α production from the $CD8^+$ T cells (Fig. 7b,e). We detected a robust TNF-α response to ZIKV-E peptide pool at 30 d.p.i. in three naïve macaques 2K2 (6.12%), 5K6 (1.7%), and 6N1 (2.7%) and in a single immune animal BS14 (3.7%). By day 60 p.i. the amount of TNF-α detected in the antigen-specific $CD8^+$ T cells was considerably diminished with only a single DENV-immune animal BS14 (0.42%) having detectable TNF-α level above the background (Fig. 7e). We detected elevated TNF-α in two naïve, 2K2 (3.5%), and 6N1 (3.0%), and one immune BS14 (3.8%) animal, in response to the ZIKV NS peptide pool; however, we were unable to detect any response above background by 60 d.p.i. of ZIKV. Interestingly,

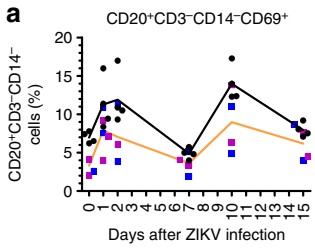 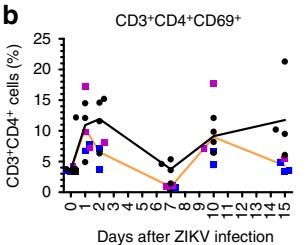 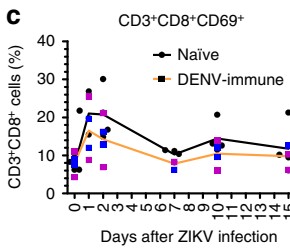

**Figure 6 | Pre-exposure to DENV results in lower activation of B and T cells.** (**a**) Frequency of activated B cells (CD20$^+$CD3$^-$CD14$^-$CD69$^+$) were statistically higher compared to their basal level in the naïve cohort at 24 and 48 h.p.i. (**b,c**) T cells were measured by flow cytometry 0, 1, 2, 7, 10 and 15 d.p.i of ZIKV. (**b**) CD4 T cells (CD3$^+$CD4$^+$CD69$^+$) showed statistically higher activation at 24 and 48 h and on days 10 and 15 p.i. in the naïve compared to their own baseline values while DENV-immune macaques had higher frequency only at 24 and 48 h p.i. Comparison of frequency of activation of B and CD4$^+$ T cells inside cohorts related to their own baseline values was performed using a two-tailed unpaired $t$-test with Sidak–Bonferroni correction. (**c**) CD8 T cell (CD3$^+$CD8$^+$CD69$^+$) subsets showing significant higher frequency in naïve cohort at 24 and 48 h and on days 7 and 10 p.i. Comparison of frequency of activation of CD8$^+$ T cells between two cohorts was performed using a two-tailed unpaired $t$-test with Sidak–Bonferroni correction. Median values are shown in orange and black lines for DENV-immune and naïve macaques, respectively. Black circles represent individual naïve macaques, and blue and magenta squares represent individuals previously exposed to either DENV-1 or DENV-2, respectively.

unlike our peptide stimulation assay, viral infection did not appear to stimulate CD8$^+$ or CD4$^+$ T cells to make TNF-α above background levels in any of the assay conditions.

CD107a production also highlight a strong antigen-specific response in the CD8$^+$ T cells, and we were able to detect it 30 d.p.i. of ZIKV, which was still present at 60 d.p.i. (Fig. 7c,f). We noticed that the DENV-immune macaques appeared to have higher percentage of CD107a$^+$ CD8$^+$ T cells in response to the stimuli (to ZIKV-E, NS1 peptides, and the whole DENV) as compared to the naïve macaques suggesting that rather than hampering the immune response, a prior DENV infection, indeed, enhanced the ability of the CD8$^+$ T cells to release lytic granules in response to their cognate antigen (Fig. 7c,f).

Akin to the CD8$^+$ T response, the ZIKV-specific CD4$^+$ T response did not appear to be impacted by prior DENV infection. We detected a robust CD4$^+$ T response in both the naïve and DENV-immune macaques 30 d.p.i. of ZIKV, which decreased by 60 d.p.i. (Fig. 7g–i). Thirty days after ZIKV infection we saw a strong polyfunctional CD4$^+$ T-cell response, where the CD4$^+$ T cells from BS14 made IFN-γ (0.70%), TNF-α (9.6%) and CD107a (2%) in response to the E peptide pool stimulation (Fig. 7g–i). As with the CD8$^+$ T response to ZIKV or DENV infection, the PBMCs induced IFN-γ and CD107a response in both the naïve and DENV-immune macaques at 30 d.p.i. of ZIKV (Fig. 7g,i), which decreased by 60 d.p.i. (Fig. 7j,l). However viral infection of the PBMCs with ZIKV or DENV again failed to induce TNF-α production (Fig. 7h,k). Nevertheless CD4$^+$ T cells from most of the naïve and DENV-immune macaques were able to induce ZIKV specific CD4$^+$ T responses after ZIKV infection suggesting that, similar to CD8$^+$ T cells, a prior DENV infection did not dampen the cellular immune response to ZIKV.

**Cytokine profile.** DENV-immune macaques showed higher basal levels of IFN-α compared to naïve macaques but they were statistically insignificant (Fig. 8a). However, IFN-α was found to be noticeably higher, compared to basal levels, only in naïve macaques at 24 ($P < 0.01$), 48 ($P < 0.05$) and 72 ($P < 0.05$) hours post infection (h.p.i.). One animal pre-exposed to DENV-2 (BS02) showed a significant increase at 24 h.p.i. By day 6 p.i., when values decreased to basal levels, naïve cohort still had more IFN-α than DENV-immune cohort ($P < 0.01$) (Fig. 8a). In terms of IFN-γ, DENV-immune macaques showed higher basal levels compared to naïve ones ($P < 0.05$). However, after ZIKV infection the level of this cytokine was not increased. Only one macaque pre-exposed to DENV-2 (BS02) had a transient surge of IFN-γ 24 h.p.i. (Fig. 8b). In contrast, all naïve macaques showed an early

pattern of IFN-γ induction with a significant increase as early as 24 h.p.i. ($P < 0.05$) without any other noticeable increase. In agreement with this early peak of IFN-γ, naïve macaques showed an early increase of the chemokine CXCL11 at 24 h.p.i. ($P < 0.01$). In contrast, DENV-immune macaques showed a decrease of CXCL11 by day 10 p.i. ($P < 0.01$) compared to baseline levels with a trend to have a sustained lower level by day 15 p.i. (Fig. 8c). ILR1-α was also increased at 24 and 48 h.p.i. ($P < 0.01$) and by days 8 and 15 p.i. ($P < 0.05$) in the naïve but not in the DENV-immune cohort. Only one macaque pre-exposed to DENV-2 (BS02) showed a transient increase in the levels of this cytokine (Fig. 8d). The T cell-activating chemokine CXCL10 was elevated in both groups at 24 h.p.i., with a significant high in naïve macaques ($P < 0.01$) and a distinct increase in DENV-immune cohort ($P < 0.05$). Thereafter CXCL10 remained high only in naïve macaques at 4 and 6 days p.i. ($P < 0.05$) (Fig. 8e). The B-cell-attracting chemokine-1 CXCL13 (BAL) level was highly variable in all four macaques in DENV-immune cohort (Fig. 8f). For the naïve cohort, this chemokine showed a slow increase p.i reaching higher levels on days 4 ($P < 0.05$), 6 ($P < 0.01$), 8 and 10 p.i. ($P < 0.05$ at both time points). B cell-activating factor (BAFF) was abundantly produced in all except in one animal (6N1) in the naïve cohort from 24 h.p.i. through 30 d.p.i. However, because of the lack of BAFF secretion by that animal the values did not reach statistically significant difference compared to baseline levels. DENV-immune cohort showed small peaks at 48 h.p.i. and by day 30 p.i. but values were not statistically relevant compared to their basal levels (Fig. 8g). Finally, level of cytolitic protein perforin increased p.i. in the DENV-immune cohort reaching statistically significant difference by 48 h.p.i. ($P < 0.05$), and maintained until 6 d.p.i. ($P < 0.05$). By days 8–30 p.i., values declined gradually (Fig. 8h). Naïve cohort produced variable levels of perforin; two macaques with high and two macaques with low levels but without significant difference compared to their baseline levels (Fig. 8h).

## Discussion

This proof-of-concept study is to advance the knowledge of the role of pre-existing flaviviral Abs during a ZIKV infection. Early work by Halstead *et al.* showed that after a secondary DENV infection viremia increased in non-human primates, suggesting that ADE may increase viral load through cross-reactive Abs. This was a limited study in macaques and only a trend was reported without showing statistically significant differences[27,28]. Despite that fact, this report has been used for decades as supporting evidence for ADE induction after heterologous DENV

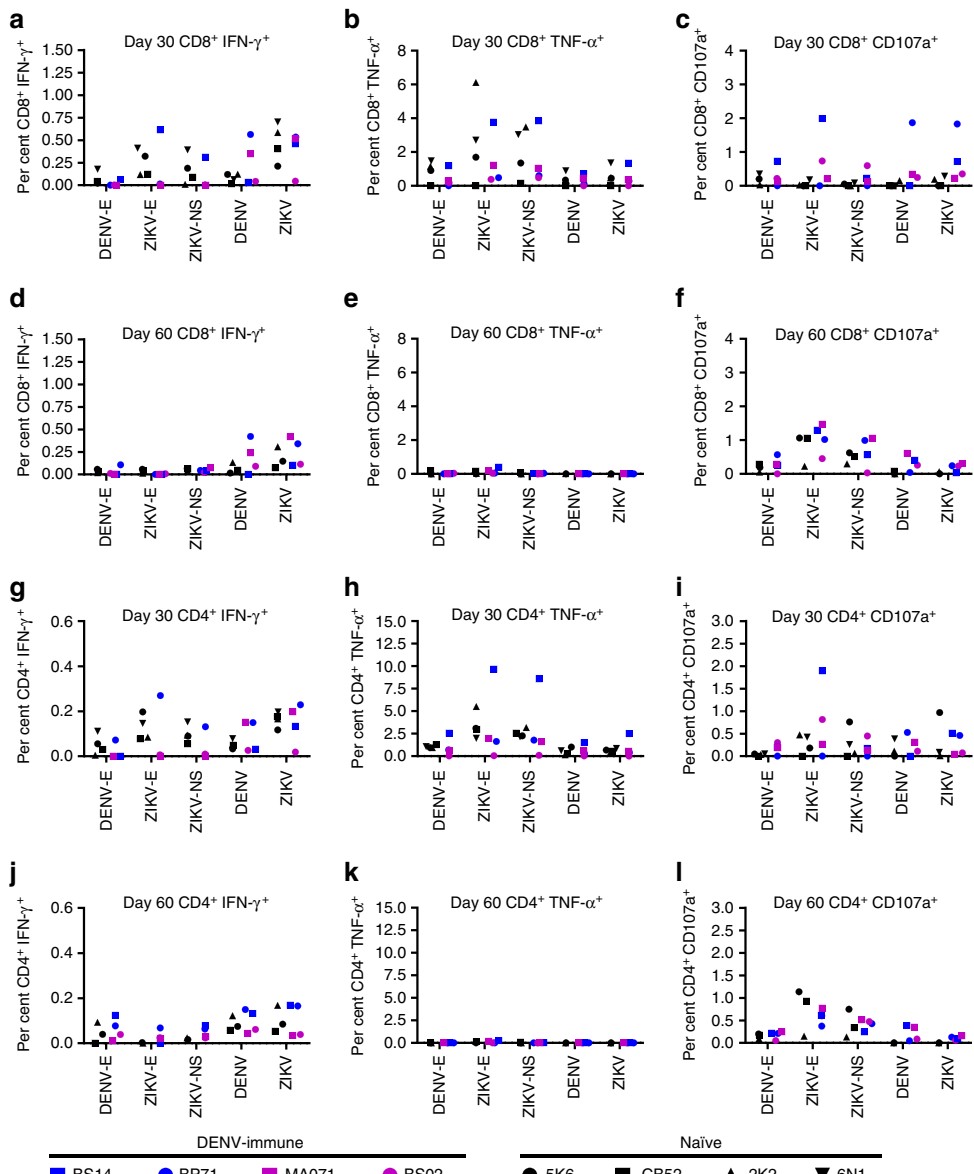

**Figure 7 | Antigen-specific CD8$^+$ and CD4$^+$ T cell responses in naïve and DENV-immune macaques.** (a–l) All percentages shown are subtracted from the unstimulated background. (a–f) Analysis of the CD8$^+$ T cell from the PBMCs of ZIKV-infected naïve and DENV-immune macaques ($n = 4$ per cohort). PBMCs were harvested on day 30 and 60 p.i., and intracellular cytokine staining of IFN-$\gamma$ (a), TNF-$\alpha$ (b) and CD107a (c) were analysed in CD8$^+$ T cells after *ex vivo* re-stimulation with the listed stimuli. Intracellular cytokine staining of IFN-$\gamma$ (d), TNF-$\alpha$ (e) and CD107a (f) were analysed for CD8$^+$ T cells after *ex vivo* re-stimulation with the listed stimuli (for naïve group 30 d.p.i. $n = 3$). (g–l) Analysis of the CD4$^+$ T cell from the PBMCs of ZIKV-infected naïve and DENV-immune macaques ($n = 4$ per group). PBMCs were harvested on day 30 and 60 p.i., and intracellular cytokine staining of IFN-$\gamma$ (g), TNF-$\alpha$ (h) and CD107a (i) were analysed in CD4$^+$ T cells after *ex vivo* re-stimulation with the listed stimuli. Intracellular cytokine staining of IFN-$\gamma$ (j), TNF-$\alpha$ (k) and CD107a (l) were analysed in CD4$^+$ T cells after *ex vivo* re-stimulation with the listed stimuli (for naïve group 30 d.p.i $n = 3$).

infection in monkeys. The concept of ADE of ZIKV after DENV infection was further reinforced by recent works that showed the ability of DENV-immune serum from humans and macaques to amplify ZIKV *in vitro* assay[15–17]. However, *in vitro* ADE assays using laboratory cell lines are notoriously promiscuous and demonstrate no correlation with disease risk. For example, DENV-immune sera will enhance even the homotypic serotype responsible for a past infection if the serum is diluted to sub-neutralizing concentrations[29]. In cell culture assays, immune sera from patients exposed to a variety of different flaviviruses including yellow fever and Japanese encephalitis viruses will enhance the infection of DENV[30]. In this study we have evaluated the potential of DENV-immune sera to enhance ZIKV using non-

human primates, which are known to be natural hosts of both viruses. We show that the long-term pre-exposure to DENV serotypes 1 or 2 induces ADE of ZIKV *in vitro,* but the *in vitro* enhancing antibodies did not increase viremia and disease when the macaques were challenged with ZIKV. Indeed, previous exposure to DENV tends to shorten ZIKV viremia. This observed trend of faster clearance of ZIKV by pre-existing dengue immunity was similar to that seen in some dengue patients after secondary infection, in which the DENV was cleared off fast[31]. In addition, we show here that previous exposure to DENV may result in modulation of the immune response induced by ZIKV without resulting in enhancement of ZIKV pathogenesis.

A recent study has demonstrated that immune sera from people exposed to DENV and West Nile virus can enhance ZIKV disease in Stat2$^{-/-}$ deficient mice[15]. We urge caution in using immune deficient mouse models to understand the pathogenesis of ZIKV in people. Both DENV and ZIKV are primate viruses that are severely restricted by the mouse type I interferon responses[32]. It is well established that in some DENV infection models, mice develop severe neurological complications and paralysis, which are rarely observed in people[12]. The threshold for antibody mediated disease in mouse models is low and antibodies have been shown to alter or enhance diseases caused by dengue, West Nile, yellow fever and Japanese encephalitis viruses in different mouse models[12,33,34]. Other than for DENVs, there is no evidence for antibodies enhancing human diseases caused by West Nile, Yellow Fever or Japanese encephalitis viruses. With respect to flaviviruses, mouse models must be judiciously used to understand well-established human phenotypes.

Previous studies have shown that the time interval between consecutive DENV infections influence inapparent versus symptomatic and severe outcome[24,25]. It has also been suggested that the window of cross-protection induced by a first infection with DENV against a second symptomatic infection is ~1–2 years[24,25]. A strength of our study is that macaques were challenged with ZIKV at 2.8 years after DENV infection when transient cross-protection is unlikely to confound the study.

The immunological characterization of the macaques used for this study is also in agreement with results observed in human populations showing that ZIKV infection in flavivirus-naïve subjects induces only low or no NAbs to other flavivirus including DENV[16,35,36]. However when ZIKV infection occurs in subjects previously exposed to other flavivirus, their samples show an expanded profile of neutralization against different flaviviruses[35,37]. In subjects with previous immunity to DENV, we show that ZIKV infection can induce a transient fourfold expansion of the Nab titres against DENV after 30 days resembling a secondary DENV infection[38]. Of interest is that after ZIKV infection, naïve macaques showed a significant increase of NAbs against DENV-4 and DENV-2, but not to the closely related DENV-1 or DENV-3 30 d.p.i. At 60 days after the infection macaques in that cohort still had NAb titres against DENV-2 and DENV4. Additional experiments including a cohort previously exposed to DENV4 and specific epitopes characterization are needed to confirm the *in vivo* meaning of these findings.

Previous results showed that Abs to NS1 obtained from memory LYM from ZIKV-infected patients were largely ZIKV-specific[39]. In contrast, one particular observation from our work is that the macaques previously exposed to DENV-2 but not to DENV-1, showed serological cross reactivity with ZIKV-NS1 as early as 30 d.p.i. of DENV. This cross-reactivity was still present in 75% of the macaques when tested 2.5 years.p.i. The region 172–352 of ZIKV-NS1 protein has been shown to be structurally similar to the NS1 structures of DENV-1 and DENV-2, and the central region of their loop surfaces have amino acid sequence identities of 53–56% (ref. 40). NS1 of DENV-1 and DENV-2 both display a positively charged surface, while NS1 of ZIKV exhibits a composite platform containing both a positively and a negatively charged central region[40]. Because the loop surface has been suggested to play a crucial role in the interactions of secreted NS1 with Abs[41], the cross reacting Abs of macaques pre-exposed to DENV-2 may be directed against this region. Another plausible explanation is that the macaques previously exposed to DENV-2, in spite of being challenged with the same amount of inoculum, had significantly more viremia days when compared to the group exposed to DENV-1, allowing for a prolonged exposure to DENV-2 NS1 protein (Supplementary Table 3, Supplementary Methods).

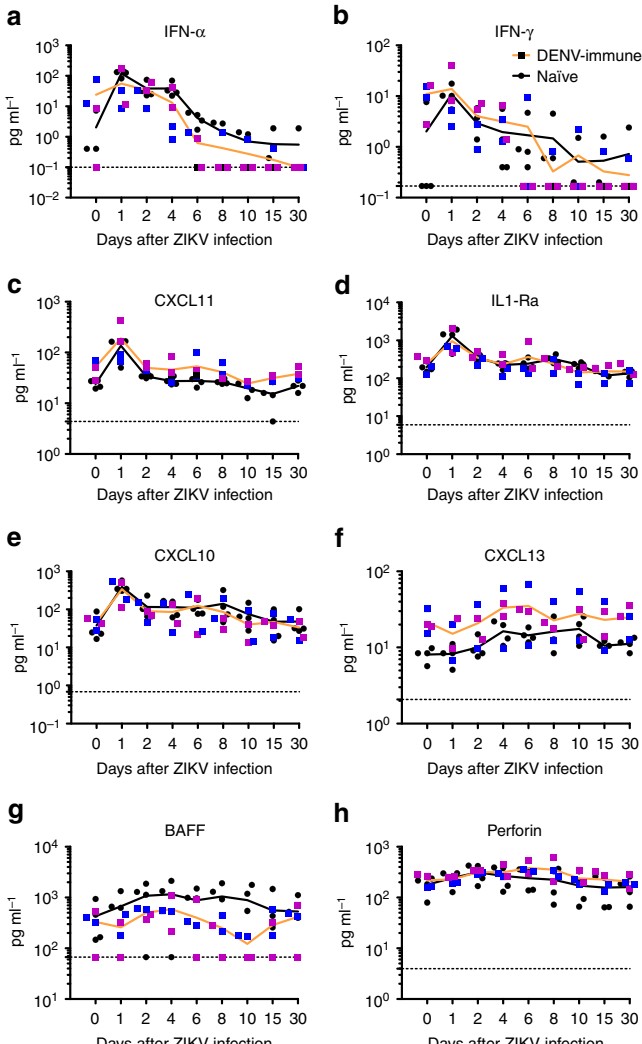

**Figure 8 | Previous exposure to DENV modulates the cytokine and chemokine profiles after ZIKV infection. (a–h)** Profiles of marked cytokines and chemokines after ZIKV infection of DENV-immune and naïve cohorts are depicted logarithmically in pg ml$^{-1}$. Mean values are shown in orange and black lines for DENV-immune and naïve cohorts, respectively. Comparison of the levels of cytokines and chemokines within cohorts related to their own baseline values or among cohorts was performed using a two-tailed unpaired *t*-test with Sidak–Bonferroni correction. Black circles represent individual naïve macaques, and blue and magenta squares represent macaques previously exposed to either DENV-1 or DENV-2, respectively. Dotted lines indicate the threshold of detection for each assay.

The detection of ZIKV viremia in the presence of existing DENV NAbs indicates that, in spite of the close relationship between ZIKV and DENV phylogenetically[35] and at structural level[4], both viruses are antigenically distant. Previously, we reported a strong negative correlation between pre-challenged Neut50 DENV titres and duration of viremia in macaques challenged with DENV after being exposed to different DENV vaccine formulations. Although some macaques with Neut50 titres >20 showed breakthrough viremia (23, 24.4, 5.5 and 28% after DENV-1, 2, 3 and 4 challenge, respectively) the viremia was shorter[11]. Furthermore, it has been shown in rhesus macaques that the serum IgG profiles and neutralization titres demonstrate that non-neutralizing, pan-reactive, and serotype-specific NAbs persist for over a year in DENV-infected macaques[42]. In this work we are reporting that 100% of the macaques with DENV

NAb titres of 1:160 or higher had ZIKV viremia and that the magnitude of that viremia did not correlate with the pre-existing DENV NAb titres. This confirms that in DENV convalescent subjects, cross-reacting Abs that are able to enhance ZIKV but neutralize DENV *in vitro* are present. However, those Abs neither have the quality to increase and extend the duration of viremia nor to abrogate ZIKV replication *in vivo*, even though the viremia tends to be shorter in the presence of dengue immunity.

Considering published data on the role of IFN-α in the induction of BAFF[43,44] the early production of BAFF observed in the naïve cohort after ZIKV infection is noteworthy when compared to the high statistically significant production of IFN-α. These results suggest the power of ZIKV to induce the innate immune response by triggering the cell pattern recognition receptors, which detect pathogen-associated molecular patterns. Previously, it was confirmed that ZIKV has the capability similar to DENV to activate the Toll-Like Receptor 3 (TLR3) pathway in human organoids and skin cells[45,46].

CXCL13 is a recognized biomarker for the activity of Follicular B helper T cells (Tfh). Plasma levels of CXCL13 strongly correlate with germinal centers (GC) Tfh-cell frequencies in draining lymph nodes after specific immunization in rhesus macaques[47]. These cells play a critical role in mediating the selection and survival of B cells capable of producing high-affinity Abs against foreign antigens, and also memory B cells capable of quick immune re-activation in the future if ever the same antigen, or related, is re-encountered[48]. Our results show a positive correlation among CD20$^+$/CD69$^+$ B cells and serum levels of CXCL13 in 75% of macaques previously exposed to DENV and having a secondary exposure to a closely related virus (ZIKV). On the other hand, 100% of the naïve macaques experiencing a first exposure to a flavivirus show statistically higher significant level of CXCL13 compared to their own basal levels. These results are confirmatory on the potential value of CXCL13 as a biological marker of GC activity and B cells activation after a primary or a secondary viral infection, in this case related to ZIKV[49]. Additional studies with large cohorts of NHP and humans are guaranteed to advance this concept.

Immune profiling of the memory CD4$^+$ and CD8$^+$ T-cell responses in the naïve and DENV-immune macaques revealed no defects in the ability of the DENV-immune cohort to respond to ZIKV infection. Notably, the DENV-1 immune animal BS14, had a strong polyfunctional CD4$^+$ and CD8$^+$ T with detectable IFN-γ and TNF-α production and a strong CD107a staining when stimulated with the ZIKV E-peptide pool, revealing that the prior DENV infection did not inhibit the ability of these macaques to support a robust response to ZIKV. We also noted that the DENV-immune macaques appeared to have higher percentage of CD107a$^+$ CD8$^+$ T cells in response to stimuli as compared to the naïve macaques, suggesting that rather than hampering the immune response, a prior DENV infection enhanced the ability of the CD8$^+$ T cells to release lytic granules in response to their cognate antigen. Further studies are needed to determine if this occurs with pre-exposure to other flaviviruses. These findings are of utmost relevance for designing ZIKV vaccines.

Resembling ZIKV infection in humans[49], ZIKV-infected macaques showed mild laboratory abnormal changes. However, we found that the variation in the absolute number and percent of LYM is more limited after ZIKV infection that has been reported after DENV infection[42]. Also of interest is the increase in liver enzymes in all ZIKV-infected macaques with a profile very similar to that reported for DENV-infected macaques[42]. However, we find that previous exposure to DENV results in a modest increase of the liver enzymes after ZIKV infection, which may suggest a protective role in liver inflammation induced by

ZIKV infection. Further studies are needed to understand the immune mechanism behind this finding.

The skin rash reported in one animal previously exposed to DENV-1 (BS14) remains inconclusive, as presence of the virus was not detected by RT-PCR in skin biopsy (Supplementary Note 1, Supplementary Table 4). The histopathology of the skin consisting of minimal amounts of non-specific lymphoplasmacytic infiltrates within the superficial dermis is similar to findings described in humans which have developed cutaneous erythematous lesions[50]. However, we consider that there is limited scientific value to a single incidence of rash.

Previous exposure to other flavivirus has been speculated to contribute to the ZIKV-associated microcephaly cases in South America, because DENV is endemic in this population. It would be interesting to know whether ADE predisposes utero infection and congenital malformations. Although that is not the focus of this study, our results provide valuable clues that the pre-existence of a long-term immune response to DENV neither enhances the peak of ZIKV viremia nor impairs the course of its infection. In addition, we show that the pre-existing immunity to DENV tends to modulate the innate, humoral and T-cell immune responses. Because fetal abnormalities have been associated with prolonged ZIKV viremia[51], and more recently, higher viremia has been linked to invasion of the central nervous system (CNS) in macaques[26], results from our work may suggest that pregnant women with previous exposure to DENV may have limited ZIKV viremia and less tendency to have invasion of CNS. These inferences invigorate the need for further direct and in-depth investigations. Our results reinforce the value of NHP model to understand the complex serological interaction among flaviviruses and to support the design of effective flavivirus vaccines.

## Methods

**Viral stock.** ZIKV H/PF/2013 strain, kindly provided by CDC, Dengue Branch, San Juan, Puerto Rico, was used in this study to provide comparative results with previously published data[52]. This ZIKV strain is known to replicate well in rhesus macaques. Virus was expanded and tittered by Plaque Assay and qRT-PCR using protocols standardized in our laboratories. DENV-1, Western Pacific 74, and DENV-2, New Guinea 44 strain used to infect macaques in 2013 and strains DENV-3 Sleman 73, and DENV-4 Dominique used for neutralization assays were kindly provided by Steve Whitehead (National Institutes of Health, Bethesda, Maryland).

**Immunizations and virus challenges of macaques.** Young adult rhesus macaques (4–5 years of age) seronegative for DENV and ZIKV were housed in the CPRC facilities, University of Puerto Rico, San Juan, Puerto Rico. All procedures were reviewed and approved by the Institute's Animal Care and Use Committee at Medical Sciences Campus, University of Puerto Rico (IACUC-UPR-MSC), and performed in a facility accredited by the Association for Assessment and Accreditation of Laboratory Animal Care (AAALAC) (Animal Welfare Assurance number A3421; protocol number, 7890115). In addition, steps were taken to ameliorate sufferings in accordance with the recommendations of the Weatherall report on The Use of Nonhuman Primates in Research. For instance, all procedures were conducted under anaesthesia by intramuscular injection of ketamine at 10–20 mg kg$^{-1}$ of body weight, as approved by the IACUC. Anaesthesia was delivered in the caudal thigh using a 23-Gauge sterile syringe needle. During the period of the entire study, the macaques were under the environmental-enrichment programme of the facility, also approved by the IACUC. Macaques were continuously monitored by trained veterinarians at the Animal Research Center. For ZIKV challenge, four macaques previously infected with DENV-1 or 2 serotype in the year 2013 (Cohort 1) and four naïve macaques (Cohort 2) were subcutaneously injected in the deltoid area with 500 μl of $1 \times 10^6$ p.f.u. of ZIKV (H/PF/2013 strain) in PBS. Both cohorts were matched in age and sex. All macaques were male and the average age for cohort 1 was 5.3 years (5.8, 5.8, 4.75 and 4.9 years) and that of cohort 2 was 5.1 years (5.8, 3.9, 5.9 and 4.58 years). While in quarantine and after ZIKV infection macaques were regularly evaluated twice daily for evidence of disease or injury. All macaques continued to eat and drink normally during this period. Weights were taken on baseline and every other day during the acute infection period (days 1–25), then at day 30 d.p.i.). Rectal and external temperature were taken daily during the acute infection period (days 1–25) and on day 30 p.i. External temperature was recorded by using an infrared

device (EXTECH Instruments, Waltham, MA). Since the macaques were already anesthetized, the infrared thermometer could be positioned close to the forehead, allowing accurate readings and constant positioning of the IR thermometer directly to the animal's forehead at a close range (within 2′) as per the manufacturer's instructions.

**In vitro ADE assay in K562 cells.** The ability of serum Abs to enhance ZIKV virus infection in K562 cells (ATCC CCL-243) was carried out with a protocol that was previously described for DENV ADE[29] with some modifications. Briefly, macaque serum was diluted fourfold starting after an initial dilution of 1:10, and then incubated with ZIKV virus (strain H/FP/2013 grown in *Aedes albopictus* cells, clone C6/36 ATCC CRL-1660) at a multiplicity of infection (MOI) of 1.0 for 1 h at 37 °C. Approximately $5 \times 10^4$ cells were added to each well in a 96 well-plate containing the mixture of the virus and the Abs. After incubation at 37 °C for 2 h, cells were washed with fresh media twice and incubated at 37 °C for 22 h. Cells were washed again, fixed with 4% PFA, permeabilized with saponin and intracellularly stained for ZIKV E protein using monoclonal antibody 4G2 conjugated to Alexa-488 diluted 1:400 (kindly provided by Dr Ralph Baric, University of North Carolina-Chapel Hill, North Carolina, USA). Per cent infection was determined by flow cytometry using a Guava flow cytometer.

**qRT-PCR.** Viral RNA for real-time PCR assay was extracted from 140 μl of virus isolate (previously tittered by plaque assay) or serum samples using QIAmp Viral RNA mini kit (Qiagen, Valencia, CA) according to the manufacturer's instructions. Real-time RT-PCR (TaqMan) assay-specific primers and probes for ZIKV were designed by Sigma-Aldrich (St Louis, MO) following the protocol developed by the Molecular Diagnostics and Research Laboratory Centers for Disease Control and Prevention, Dengue Branch at San Juan, PR[35,53]. Primers and probe sequences are provided in Supplementary Table 5. RNA from other flaviviruses were included as negative control. For the reaction mixture, 5 μl of RNA was combined with 100 μM primers and 25 μM probe in a 25 μl total volume using Life Technologies SuperScriptIII Platinum assay kit (Life Sciences). Assays were performed in an iCycler iQ Real Time Detection System (Bio Rad, CA). For quantification, a standard curve was generated from tenfold dilutions of RNA from a known amount of virus, the titre of which was determined by plaque assay as previously described[54–56].

**ELISA for DENV and ZIKV.** Before conducting ZIKV infection studies, we characterized the macaques' immune response against DENV and ZIKV. For this study, we used acute and convalescent samples (1 and 2.5 years after DENV infection) collected from all macaques infected with DENV-1 ($n = 4$) or DENV-2 ($n = 4$) 2.8 years before ZIKV infection. From all four DENV-infected macaques we selected two macaques per serotype to expose to ZIKV. Seroreactivity to DENV was tested using commercial IgG and IgM ELISA kits (Focus, Cypress, CA). Anti ZIKV-NS1 IgG was examined using a commercial kit (Alpha Diagnostic, San Antonio, TX), and ZIKV IgM was assessed with an available commercial kit (InBios, Seatle, WA).

**DENV and ZIKV titration and FRNT60/PRNT60.** Assay was performed by seeding Vero81 cells (ATCC CCL-81) ($\sim 8.5 \times 10^4$ per well) in 24 well plates with DMEM (Dulbecco's Modified Eagle's medium, Thermo Fisher Scientific). The next day tenfold dilution of the virus was made in diluent medium (Opti-MEM (Invitrogen) with 2% FBS (Gibco) and 1% antibiotic/antimycotic (HyClone). Prior to inoculation, growth medium was removed and the cells were inoculated with 100 μl per well of each dilution in triplicates, and the plates were incubated 1 h/37C/5%CO2/rocking. After incubation, the viral dilutions were overlaid with 1 ml Opti-MEM including 1% Carboxymethylcellulose (Sigma), 2% FBS, 1% non-essential aminoacids (Gibco), 1% antibiotic/antimycotic (HyClone). After 3–5 days of incubation at 37 °C/5%CO2, overlay was removed; the cells were washed twice with phosphate buffered saline (PBS) and fixed in 80% methanol.

For ZIKV, cells were stained with crystal violet. For DENV, plates were blocked with 5% non-fat dry milk in PBS and incubated for 1 h per rocking with anti-E mAb 4G2 and anti-prM mAb 2H2 (kindly provided by Dr Aravinda de Silva), both diluted 1:250 in blocking buffer. The plates were then washed and incubated 1 h with horseradish peroxidase (HRP)-conjugated goat anti-mouse antibody (Sigma), diluted 1:1,000 in blocking buffer. Foci were developed with TrueBlue HRP substrate (KPL) and counted. For the Focus/Plaque Reduction Neutralization Test (FRNT/PRNT), sera were diluted twofold and mixed with $\sim 35$ focus per plaque-forming units (FFU per p.f.u. per ml) of virus and then incubated for 1 h at 37 °C. Virus-serum dilutions were added to the cells and incubation was continued for 1 h at 37 °C per rocking, and overlay medium was added and processed as described above. Mean focus diameter was calculated per $\geq 20$ foci per clone measured at $\times 5$ magnification. Results were reported as the FRNT or PRNT with 60% or greater reduction in DENV or ZIKV focus or plaques (FRNT60 or PRNT60); and 1:20 is considered a positive titre, and <1:20 is considered a negative Neut titre.

**Binding assay.** For virus capture ELISA assay, Microlon 600 96-well plates (VWR, Radnor, PA) were coated overnight at 4 °C with a mixture of 100 ng of 4G2/2H2 Abs in 0.1 ml coating buffer (Sigma, 08058). Unbound Abs were washed off with PBS containing 0.05% Tween 20, and blocked with 3% BSA (Fisher). Virus was diluted in blocking buffer to optimal concentrations, and 0.1 ml was added to each well and incubated 1 h at 37 °C. Unbound viruses were removed by washing, and serum samples diluted in blocking buffer were added and incubated for another hour. Unbound serum was removed by washing and then incubated for 1 h at 37 °C with goat-anti-monkey secondary Ab conjugated with HRP (Fitzgerald, MA, USA). Unbound secondary Ab was washed off, and signals were developed with o-phenylenediamine dihydrochloride substrate tablets (Sigma, 34006). OD was read at 492 nm.

**Phenotyping of immune cells.** Phenotypic characterization of rhesus macaque PBMCs was performed by multicolour flow cytometry using direct immunofluorescence. Aliquots of 150 μl of heparin whole blood were directly incubated with Abs for 20 min. at room temperature; red blood cells were lysed with ACK, and cells were then washed twice with PBS and fixed with 1.6% methanol-free formaldehyde before analysis in a CyAn LDP (Beckman-Coulter, USA) flow cytometer. Antibodies used in this study were: CD123w-APC (7G3), CD3 FITC or v500 (SP34 and SP34.2, respectively), CD4-PerCP-Cy5.5 (L200) and HLA-DR PerCP-Cy5.5 (G46.6) from BD-Biosciences; CD14-FITC (322A-1 (My4)), CD159a (NKG2A)-PE (Z199), CD20-FITC (H299(B1)), CD335 (NKp46)-PC7 (BAB281) and CD337 (NKp30)-PC5 (Z25) from Beckman-Coulter; CD11c-PE (3.9), CD16-Alexa Fluor 700 (3G8) and CD20-APC (2H7) from Biolegend; CD69-PE (FN50) from DAKO, CD8-Pacific blue (3B5) from Invitrogen; CD66abce-FITC (TET2) from Miltenyi Biotec. and CD159c (NKG2C)-APC (134522) from R&D Systems. Information on dilution of Abs is provided in Supplementary Table 6. For analyses, LYM were gated based on their characteristic forward and side scatter pattern; T cells were then selected with a second gate on the CD3 positive population. CD8 T cells were defined as $CD8^+CD3^+$ and CD4 T cells were $CD4^+CD3^+$. Natural Killer cells were defined as $CD3^-CD20^-CD14^-$ and analysed by the expression of NK cell markers CD16, CD8, NKG2A, NKG2C, NKp30 and NKp46. B cells were defined as $CD20^+CD3^-CD14^-$ (Supplementary Table 7). The activation marker CD69 was determined in each different lymphoid cell population (Supplementary Fig. 5). Monocytes were defined as $CD20^-CD3^-CD14^+$ and $CD20^-CD3^-CD14^+CD16^+$. Finally, dendritic cells (DCs) were separated in two populations by the expression of CD123 (pDCs) or CD11c (mDCs) in the $HLA-DR^+$ $CD3^-$ $CD14^-$ $CD20^-$ $CD66^-$ population.

**Cell-immune response assessment.** Intracellular cytokine staining of immune rhesus macaque PBMCs was performed by multicolour flow cytometry using methods similar to those described by Meyer *et al.*[57]. Detailed information of the used Abs is provided in Supplementary Table 8. Briefly, $\sim 1 \times 10^7$ day 30 and day 60 PBMC samples from both the naïve and DENV-immune macaques were thawed 1 day prior to stimulation. Each sample was aliquoted and $1.5 \times 10^6$ PBMCs were infected overnight with a combination of DENV-1 (WP 74), and DENV-2 (NGC44) at a MOI of $\sim 0.1$ or ZIKV at a MOI of 0.5 in RPMI medium with 5% FBS. The remaining $7 \times 10^6$ PBMCs were rested overnight as described earlier[58] in 5 ml of RPMI with 10% FBS. These rested PBMCs were then stimulated for 6 h at 37 °C, 5% CO2 with α-CD3-1 (100 ng ml$^{-1}$), ZIKV-E peptides (15-mers overlapping by 10 amino acids, 2.5 μg ml$^{-1}$ per peptide), ZIKV-NS1 protein peptides (15-mers overlapping by 10 amino acids, 475 ng ml$^{-1}$ per peptide), or DENV-1 and DENV-2 E peptides (1.25 μg ml$^{-1}$), all in the presence of brefeldin A (10 μg ml$^{-1}$), α-CD107a-FITC (H4A3) (10 μl), and co-stimulated with α-CD28.2 (1 μg ml$^{-1}$) and α-CD49d (1 μg ml$^{-1}$). Peptides sequences used on this experiment are provided as Supplementary Table 9. After stimulation, the cells were stained for the following markers: CD4-PerCP Cy5.5 (Leu-3A (SK3), CD8b-Texas Red (2ST8.5H7), CD3-PacBlue (SP34), CD20-BV605 (2H7), CD95-V510 (DX2), CD28.2-PE-Cy5, IFN-γ-APC (B27) and TNF-α-PE-Cy7 (MAB11). The samples were then run on an LSRII (BD) and analysed using Flowjo (Treesar). Lymphocytes were gated based on their characteristic forward and side scatter pattern, T cells were selected with a second gate on the CD3-positive population, and at the same time CD20-positive cells were excluded.

CD8$^+$ T cells were defined as $CD3^+CD20^-CD8^+$ and CD4$^+$ T cells as $CD3^+$ $CD20^-$ $CD4^+$. Cytokine expression was determined by the per cent $CD4^+$ or $CD8^+$-positive cells, and then stained positive for the cytokine IFN-γ or TNF-α. CD107a were also measured in these populations to determine functional cytotoxicity. Further analysis was also performed to examine CD28 and CD95 expression on the LYM populations to study the presence of central and effector memory cell populations.

**Multiplex cytokine analysis of plasma.** Rhesus macaques serum samples were analysed for 32 cytokines and chemokines by Luminex using established protocols for Old World primates[59]. Evaluation of analytes B lymphocyte chemoattractant (BLC, CXCL13), B cell-activating factor (BAFF), eotaxin (CCL11), GRO a (CXCL1), interferon alpha (IFN-a), IFN-g, interleukin-1 beta (IL-1b), IL-1 receptor antago nist (IL-1RA), IL-2Ra, IL-4, IL-6, IL-8, IL-10, IL-12 p70, IL-18, IL-22, IL-23, interferon gamma-induced protein 10 (IP-10, CXCL10), interferon-inducible T-cell

alpha chemoattractant (I-TAC, CXCL11), monocyte chemoattractant protein 1 (MCP-1, CCL2), macrophage migration inhibitory factor (MIF), macrophage inflammatory protein 1-alpha (MIP-1a, CCL3), MIP-1b (CCL4), perforin, regulated on activation, normal T cell expressed and secreted (RANT ES, CCL5), tumour necrosis factor-alpha (TNF-a TNF-b, and soluble CD40 ligand (sCD40L) were included in this assay. Only analytes quantifiable above the limit of detection are presented.

**Statistical methods.** Statistical analyses were performed using GraphPad Prism 6.0 software (GraphPad Software, San Diego, CA, USA). For viral burden analysis, the log titres and levels of vRNA were analysed by one-way ANOVA. The statistical significance between or within groups evaluated at different time points was determined using two-way analysis of variance (ANOVA) (Bonferroni's multiple comparisons test) or unpaired $t$-test to compare the means. The $P$ values are expressed in relational terms with the alpha values. The significance threshold for all analyses was set at 0.05; $P$ values less than 0.01 are expressed as $P < 0.01$.

**Data availability.** All relevant data are available from the authors upon request.

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

## Acknowledgements

We thank the staff of the Caribbean Primate Research Center and the Animal Resources Center. We thank Laura M. Parodi for phenotyping of immune cells and multiplex cytokine analysis, and thank Mr Ian Cacho Santana for his technical support during the preparation of this manuscript. We also acknowledge American Science and Medical Writers (Fairlawn, OH) for editorial assistance in preparation of this manuscript. This work was supported by grants 2 P40 OD012217 and 2U42OD021458-15 (OD, NIH) to M.I.M. and C.A.S. Partial support was also provided by grants K22AI104794 (NIAID) to J.D.B., G12MD007600 (RCMI) to P.P., and NIGMS-RISE Program (R25GM061838) of UPR-Medical Sciences Campus to E.X.P.-G.

## Author contributions

C.A.S., P.P. and E.X.P.-G. designed the experiments. C.A.S., E.X.P.-G., P.P., L.J.W. and A.d.S., analysed the results and drafted the manuscript. P.P., E.X.P.-G., L.C. prepared viral stocks and performed plaque assays. P.P. and E.X.P.-G. performed viral load assays. P.P. and C.S. developed and performed binding assay. L.G. and V.H. performed immunophenotyping and cytokines assays, produced and analysed the results and drafted the specific section in the manuscript. M.A.H., J.D.B., A.K.P. designed and performed the T-cell immune response characterization, produced and analysed the results and drafted that section in the manuscript. I.V.R. and M.I.M. supervised the protocol working with macaques. I.V.R. supervised and executed the macaques experiments and coordinated macaques samples collection. T.A. and L.C. performed the enzyme immune assays and coordinate the samples distribution. O.G. performed the pathology work. The publication's contents are solely the responsibility of the authors and do not necessarily represent the official views of Office of the Director, National Institutes of Health.

## Additional information

**Competing interests:** The authors declare no competing financial interests.

