## [Peer Review File · Nature Communications]

Reviewers' comments:

Reviewer #1 (Remarks to the Author):

The study objective was to determine whether DENV-immune macaques would develop enhanced viremia in the context of a ZIKV infection. This question addresses whether the phenomenon of antibody dependent enhancement (ADE), which has been observed with a secondary DENV infection of a different serotype, can amplify the viremia associated with a ZIKV infection. ADE has been speculated to contribute to the concentration of ZIKV-associated microcephaly cases in South America, because DENV is endemic in this population and most adults have immunity to DENV. Contrary to the study hypothesis, the authors found that DENV-immune macaques had fewer viremic days and lower levels of many cytokines associated with acute ZIKV infection. The study findings are very interesting, but results should be considered preliminary given the modest number of macaques in the study and the authors should be careful regarding their level of speculation about their results. However, the lack of a dramatic ADE response in vivo is an important finding. Overall, the results and discussion were very difficult to follow and should be presented in a more focused and integrated manner for the reader. I think it would be clearer to present the ADE in vitro results first, prior to explaining the study design and then following with the clinical results in the macaques. Please exclude the "highly suspicious" results for vRNA in saliva as the level of detection for the assay was not met. A native English speaker should proofread the manuscript for errors (grammatical, spelling, and wrong verb choice), which were distracting and confusing at times. Overall, the results are quite interesting, but do not meet criteria for publication in an apex journal.

Reviewer #2 (Remarks to the Author):

In this manuscript, Pantoja-Maldonado et al report on the outcome of ZIKV infection in rhesus macaques that had been infected with DENV-1 or DENV-2 approximately 3 yrs earlier in comparison to DENV-naïve animals. The authors performed detailed clinical, virological, and immunological analyses. The authors found that sera of DENV-immune animals enhanced ZIKV infection in vitro without detectable anti-ZIKV neutralization. Following ZIKV challenge, there were minor differences in clinical hematology results, a trend toward less elevation of AST and ALT in DENV-immune animals, and similar peak viremia levels with a non-significant trend towards shorter viremia duration in DENV-immune animals. Cytokine levels and frequencies of activated lymphocytes tended to be lower in DENV-immune animals with the exception of serum CXCL13 and perforin levels.

Published data demonstrating enhancement of ZIKV infection in vitro by DENV-immune sera along with the known association of severe dengue illness with secondary infection and the co-circulation of DENV and ZIKV have raised concerns about a detrimental interaction between DENV and ZIKV. Therefore, the results of the present study would likely be received with great interest by the scientific community, despite the relatively small sample sizes and consequently limited statistical power. Since the interval between infections is known to have an important influence on the outcome of secondary DENV infection, the availability of these animals infected with DENV several years earlier represents a unique strength to the present study that otherwise might not be duplicated for several years.

However, the manuscript is exceedingly poorly written and contains a number of technical flaws that greatly detract from its scientific merit.

Major comments:

1. The manuscript requires careful proofreading and extensive editing. Specific examples of

typographical errors and poor sentence construction are too numerous to catalog. In many instances these render the intended meaning unclear. In several places, the text references an incorrect figure or table.

2. The manuscript does a poor job of highlighting key findings. The Abstract should be substantially revised to better describe the studies done and the main findings, whereas insignificant findings should be mentioned briefly if at all. The Results section can be significantly shortened, referring to tables and figures rather than repeating all findings, particularly minor or insignificant ones. The Discussion should be more focused on placing the major findings in context.

3. The manuscript does not adequately address current knowledge on sequential flavivirus infections. In particular, there is no mention of the important effect of interval between infections, which should be highlighted in the Abstract and Discussion as it is a unique feature of the present study. Also, there is a substantial literature on cross-serotype protection (against disease) in sequential DENV infections that needs to be noted and discussed in the context of the findings presented.

4. Statistically significant results need to be presented more clearly. The comparisons tested should be clearly stated whenever p values are presented- for example, in some cases, the authors appear to be comparing different time points in the same animals rather than different groups. The statistical tests used should also be noted.

5. The flow cytometry data are problematic as currently presented. Fluorochromes, antibody panels, and cytometer configuration need to be noted. Representative flow cytometry data illustrating the gating strategy and definition of subsets analyzed should be included. Monocytes are described as CD3+ in the text and figures, which is not appropriate. Activation marker expression should be presented as a percentage of each cell population rather than as a percentage of total PBMC, and the distribution of cell subsets within PBMC should be included. Figure 4e appears to indicate that NK cells frequently constituted >50% of PBMC (in some samples, >80%); this is unexpected and should be addressed. Day 7 results of CD69 staining appear to be an outlier, and raise concern about technical problems with these samples; the authors should address quality control used to ensure that these results are valid. The timing of B and T cell activation observed is unusual and surprising, with significant changes at days 1 and 2 post-infection; the authors' statement that the "pattern of activation ... were similar to those previously reported" is not justified from the article cited.

6. In several places in the manuscript, the authors refer to virus-specific "total IgG". It is not clear how this was defined; ELISA OD values are only indirect measures of total IgG and this distinction should be kept clear throughout. The discussion of "fold increases" in OD values is not a good measure of antibody responses, in particular for comparing flavivirus-naïve and flavivirus-immune groups, and should be deleted.

7. Several hematologic and immunologic parameters showed significant differences between DENV-immune and DENV-naïve animals at baseline (day 0), e.g., neutrophils, IFN-alpha, CXCL13. This should be addressed as a potential confounding factor.

Minor comments:

8. Abstract- The conclusion that CXCL13 might be useful as a biomarker of acute ZIKV infection is not warranted. The reference to "in response to stimuli" for CD8/CD107a cells should be clarified. The Abstract should end with a conclusion.

9. Author summary- The authors should clarify the meaning of "results in severe forms".

10. Introduction- The concluding paragraph should be revised to more specifically address the objectives of the present study.
11. Page 6 line 3- "quarantine period" is not explained in the Methods section.
12. Page 6 line 22- Re: "significant" decrease in neutrophils, it appears that baseline neutrophil counts were higher in this group; this should be noted in the text as a possible explanation for the difference.
13. Page 6 line 27- Supplemental figure 4f does not show an increase in CD14+CD16+ cells as described in the text.
14. Page 9 line 4- The meaning of this sentence ("titers for anti ZIKV-NS1 protein were also expanded") is unclear.
15. Page 9 lines 29, 32- The use of "endpoint" is incorrect in these sentences ("higher endpoint dilution titers in the first two dilutions").
16. Page 10 line 11- The meaning of "highest enhancement potential" is unclear.
17. Page 10- Given the small sample sizes, it would be preferable to express frequencies (of animals with positive PCR assays) as fractions (e.g., "2 of 4") rather than percentages; the latter are misleading.
18. Page 10-11- The low yield of PCR testing on urine is unexpected and worthy of comment by the authors.
19. Page 11 line 21-24 and page 14 lines 23-25- It is not clear what data justify the conclusion that there was greater variability in results in the DENV-immune group than in the naïve group. The Discussion of this point (page 19) should be modified.
20. Page 12-13- The range of cytokine/CD107a frequencies in unstimulated background control samples should be noted.
21. Page 14 line 3- The data in figure 7c and 7f do not appear to justify the statement that CD8+ T cell CD107a responses were diminished on day 60 relative to day 30.
22. Page 17 lines 5-6- This statement ("occurs only in about 1% of human population") is inaccurate; it refers to severe dengue illness rather than ADE.
23. Page 18- Can the authors suggest alternate hypotheses for the finding of anti-ZIKV NS1 in DENV-2-immune animals but not DENV-1-immune animals? For example, could differences in the content of NS1 protein in the DENV inocula explain this result?
24. Page 19-20- The authors have not presented data to claim a "perfect correlation" between BAFF and IFN-alpha production (page 19 line 16) or a "positive correlation" between CD20+/CD69+ B cells and serum levels of CXCL13 (page 20 line 5).
25. Page 20 lines 29-30- It is not clear what the authors intend by this statement ("This confirms the relevance..."). They have not presented evidence that lower viremia in DENV-immune animals (which

was not statistically significant) is related to cell-mediated immunity.

26. Page 21- There is limited scientific value to a single instance of rash; the Discussion of this point should be significantly curtailed.

27. Page 21 lines 22-23- The authors have not presented any data on "NK activity specific for ZIKV".

28. Page 21 ("we are establishing that the presence of previous immunity to DENV shortens infection")- Since results were not statistically significant, the authors should be more cautious in their conclusion.

29. Animal cohorts (page 26)- It is particularly important that the authors describe the animal cohorts more clearly. The authors should be more specific regarding matching for age. Figure 1 is not particularly helpful in showing the timeline of events and should be revised.

30. Page 26 line 26- This presumably should read "seronegative for DENV and ZIKV".

31. qRT-PCR (page 27)- The limit of quantitation should be defined for the qRT-PCR assay used. Further detail should be provided to justify inclusion of data on "highly suspicious positive" assay results, particularly evidence that these are not false positive results.

32. PRNT/FRNT assay (page 28)- The authors should clarify the use of mean focus diameter, whether the reduction measured was in focus/plaque diameter or number, and whether titers reported are endpoint titers or calculated using other methods. (Based on the values reported, it is likely that these are endpoint titers.)

33. Immune response assays (page 30)- The peptides used should be described in additional detail.

34. Figure 2b- The y axis scale should be revised, e.g., 90 to 105 degrees F. It would be considered more appropriate scientifically to express temperature in degrees C. Also, rectal temperatures (supplemental figure 1a) would be preferable to include in the main results in place of figure 2b.

35. Figure 3 duplicates data from Table 1b; it is unnecessary to present these data in both table and graph formats. Figure 3c does not add to the manuscript and can be eliminated.

36. Figure 4- The y axes in panels a and b should be labeled "OD".

37. Figure 4c- The x axis should not use both logarithmic notation and a label stating "log₁₀ reciprocal value").

38. Figure 5a duplicates data presented in Table 1a; it is unnecessary to present these data in both table and graph formats.

39. Figure 5b- The y axis is confusing; the major tick labels are linear but logarithmic minor tick marks are shown. This figure should adopt a standard logarithmic scale without breaks as in Figure 5a.

40. Figure 7- The corresponding panels for day 30 and day 60 responses should utilize the same y axis range to facilitate comparisons.

41. Table 1- Tables 1a and 1b present different information and should be separated.

42. Supplemental figure 2- The micrographs shown need better description of the specimen and the staining method.

43. Supplemental figure 3- Since OD values are already presented in panels c and e, the information in panel f is unnecessary.

Reviewer #3 (Remarks to the Author):

Pantoja-Maldonado et al. reported on pre-existing dengue immunity does not affect Zika pathogenesis but modulates the immune response in rhesus macaques. This subject is an important issue to the field of flaviviruses for many decades, especially for dengue virus (DENV) and Zika viruses (ZIKV) infections. The current well-designed investigations and in depth analysis of the insignificant effects of pre-existing dengue immunity on ZIKV infection provide an essential message on the argument of the role of in vitro artifact of antibody dependent enhancement (ADE) in disease pathogenesis induced by flaviviruses. The authors singled out critical information on the small number of rhesus macaques utilized in early days to investigate the ADE, coupling mainly with in vitro experimental studies, the hypothesis since then has become a legend, but constantly being debated and challenged. The current in vivo data will surely likely put out the hypothesis mainly derived from in vitro in peace to the field of DENV and ZIKV infections. One of the worth mentioning is that the observed profile on the faster clearance off ZIKV in pre-existing dengue immunity was similar to that of seen in dengue patients in secondary infection, in which the DENV was cleared off faster in secondary infection 1. As such, the in vivo information provided in the current manuscript will make a giant step forward to the pathogenic causes derived from DENV and ZIKV infections, so do vaccine development and control.

Minor comments:

1. The authors mentioned during quarantine period; Did the quarantine come after the DENV infection or naive animals prior to DENV followed by ZIKV infections?
2. Measuring the body weights of rhesus monkeys required to put down the animals, will the procedure effect on the reading and the observed outcomes reported here?
3. The temperatures were measured with an infrared device, how accurate the readings were? Were the readings affected by distance and angle of the measurements?
4. Despite the platelet counts remained constant, was the functionality of the platelets the same?
5. With the notion that asymptomatic infection and latent of DENV infection may occur in rhesus monkeys, hence, hence the animal with rash may shed DENV. It may be helpful to sort out whether DENV vRNA was in the collected specimens that were tested samples for ZIKV vRNA?

1. Vaughn, D.W., et al. Dengue in the early febrile phase: viremia and antibody responses. J Infect Dis 176, 322-330 (1997).

Answers to the reviewers' comments

Reviewer #1 (Remarks to the Author):

The study objective was to determine whether DENV-immune macaques would develop enhanced viremia in the context of a ZIKV infection. This question addresses whether the phenomenon of antibody dependent enhancement (ADE), which has been observed with a secondary DENV infection of a different serotype, can amplify the viremia associated with a ZIKV infection. ADE has been speculated to contribute to the concentration of ZIKV-associated microcephaly cases in South America, because DENV is endemic in this population and most adults have immunity to DENV. Contrary to the study hypothesis, the authors found that DENV-immune macaques had fewer viremic days and lower levels of many cytokines associated with acute ZIKV infection. The study findings are very interesting, but results should be considered preliminary given the modest number of macaques in the study and the authors should be careful regarding their level of speculation about their results. However, the lack of a dramatic ADE response in vivo is an important finding.

We appreciate the reviewer's observations and suggestions. We have revised the manuscript and discussed our results the way they are in order to be more objective. Although we agree in part with the reviewer's judgment that our results are preliminary due to the fact that modest number of macaques were used, having even a modest number of animals exposed to a flavivirus in a controlled manner and maintained for two half years is an asset to the biology of ZIKV in the context of a fast-spreading pandemic. As the reviewer has mentioned these findings are of high relevance to understand the role of pre immunity to dengue in the outcome of Zika infection.

I think it would be clearer to present the ADE in vitro results first, prior to explaining the study design and then following with the clinical results in the macaques.

We appreciate the reviewer's suggestion. We have moved the ADE in vitro results to the top in the Results section.

Please exclude the "highly suspicious" results for vRNA in saliva as the level of detection for the assay was not met.

As per the reviewer's suggestion, qRT-PCR results from saliva samples have been excluded.

Overall, the results and discussion were very difficult to follow and should be presented in a more focused and integrated manner for the reader.

A native English speaker should proofread the manuscript for errors (grammatical, spelling, and wrong verb choice), which were distracting and confusing at times.

As per this reviewer's and others' suggestion, the manuscript has been extensively edited and rewritten by a native English speaker and science editor. The language and grammar have been vastly improved, the whole manuscript has been rewritten for better clarity and presentation, and the manuscript is correctly formatted to make it fit for publication in Nature Communications.

Answers to the reviewers' comments

Overall, the results are quite interesting, but do not meet criteria for publication in an apex journal.

We humbly disagree. Our results are timely, invaluable to the understanding of ZIKA pathogenesis, capable of eliminating long-held misconceptions, and will strongly impact on all scientific communities and populations dealing with dengue, Zika, and other flaviviruses who are desperately searching for answers. Therefore, we strongly believe that our results deserve to be published in a widely noticeable high-impact journal such as Nature Communications.

Reviewer #2 (Remarks to the Author):

In this manuscript, Pantoja-Maldonado et al report on the outcome of ZIKV infection in rhesus macaques that had been infected with DENV-1 or DENV-2 approximately 3 yrs earlier in comparison to DENV-naïve animals. The authors performed detailed clinical, virological, and immunological analyses. The authors found that sera of DENV-immune animals enhanced ZIKV infection in vitro without detectable anti-ZIKV neutralization. Following ZIKV challenge, there were minor differences in clinical hematology results, a trend toward less elevation of AST and ALT in DENV-immune animals, and similar peak viremia levels with a non-significant trend towards shorter viremia duration in DENV-immune animals. Cytokine levels and frequencies of activated lymphocytes tended to be lower in DENV-immune animals with the exception of serum CXCL13 and perforin levels.

Published data demonstrating enhancement of ZIKV infection in vitro by DENV-immune sera along with the known association of severe dengue illness with secondary infection and the co-circulation of DENV and ZIKV have raised concerns about a detrimental interaction between DENV and ZIKV. Therefore, the results of the present study would likely be received with great interest by the scientific community, despite the relatively small sample sizes and consequently limited statistical power. Since the interval between infections is known to have an important influence on the outcome of secondary DENV infection, the availability of these animals infected with DENV several years earlier represents a unique strength to the present study that otherwise might not be duplicated for several years.

We can't thank this reviewer enough for vetting our manuscript thoroughly, noticing the strengths of our results, quoting excellent observations, highlighting weaknesses, and giving us a very comprehensive help to improve the manuscript to ensure suitability for publication in Nature Communications. It's been a pleasure to answer all forty-three comments with our best effort possible.

We also acknowledge this reviewer for highlighting the uniqueness of the cohort used in our studies.

Major comments:

However, the manuscript is exceedingly poorly written and contains a number of technical flaws that greatly detract from its scientific merit.

Answers to the reviewers' comments

1. The manuscript requires careful proofreading and extensive editing. Specific examples of typographical errors and poor sentence construction are too numerous to catalog. In many instances these render the intended meaning unclear. In several places, the text references an incorrect figure or table.

We apologize for our oversight. Reviewer #1 also has raised the same concerns. As per the suggestion of these reviewers, the manuscript, including Figure and Table legends, has been extensively edited and rewritten by a native English speaker and science editor. We have made every effort to make it fit for publication in Nature Communications. Please also see our answer above to Reviewer#1.

2. The manuscript does a poor job of highlighting key findings. The Abstract should be substantially revised to better describe the studies done and the main findings, whereas insignificant findings should be mentioned briefly if at all.

We appreciate this observation. The manuscript has been extensively revised. Now only the main findings are highlighted.

The Results section can be significantly shortened, referring to tables and figures rather than repeating all findings, particularly minor or insignificant ones. The Discussion should be more focused on placing the major findings in context.

We thank you for the suggestion. Both Results and Discussion sections were rewritten and shortened to be more focused on the main results.

3. The manuscript does not adequately address current knowledge on sequential flavivirus infections. In particular, there is no mention of the important effect of interval between infections, which should be highlighted in the Abstract and Discussion, as it is a unique feature of the present study.

We agree with the reviewer that highlighting the role of the interval between infections will improve the value of the work described in this manuscript. In the original version we thought we mentioned this. However, after seeing the reviewer's comment this has been better addressed in the revised version. Sentences have been added to the Abstract, Introduction, and Discussion sections.

Also, there is a substantial literature on cross-serotype protection (against disease) in sequential DENV infections that needs to be noted and discussed in the context of the findings presented.

Due to the limited number of literature that is allowed to be cited (Journal format) we made reference to review articles on this topic more than to specific work on this subject. Here we are including highly important references on cross-serotype protection and the role of the maturity of the immune response to a previous dengue exposure. At least four new works have been cited on this topic:

Answers to the reviewers' comments

- Montoya, M., *et al.* Symptomatic versus inapparent outcome in repeat dengue virus infections is influenced by the time interval between infections and study year. *PLoS Negl Trop Dis* **7**, e2357 (2013).
- Anderson, K.B., *et al.* A shorter time interval between first and second dengue infections is associated with protection from clinical illness in a school-based cohort in Thailand. *J Infect Dis* **209**, 360-368 (2014).
- Endy, T.P., *et al.* Relationship of preexisting dengue virus (DV) neutralizing antibody levels to viremia and severity of disease in a prospective cohort study of DV infection in Thailand. *J Infect Dis* **189**, 990-1000 (2004).
- Guzman, M.G., *et al.* Enhanced severity of secondary dengue-2 infections: death rates in 1981 and 1997 Cuban outbreaks. *Rev Panam Salud Publica* **11**, 223-227 (2002).

4. Statistically significant results need to be presented more clearly. The comparisons tested should be clearly stated whenever p values are presented- for example, in some cases, the authors appear to be comparing different time points in the same animals rather than different groups. The statistical tests used should also be noted.

The reviewer is right. In some instances, we made intragroup comparisons more than making comparison between two cohorts. This has been clarified in the text. Statistical tests have been noted in each needed case. Also we have clarified when we compare different time points in the same group and when in different groups. The following text was added for clarity

Figure 3: Comparison of absolute neutrophils numbers within cohorts related to their own baseline values was performed using a two-tailed unpaired t test with Sidak-Bonferroni correction.

Similar statements have been included in Figs , 4, 6, and 8 , and Supplementary data Fig 1.

5. The flow cytometry data are problematic as currently presented. Fluorochromes, antibody panels, and cytometer configuration need to be noted. Representative flow cytometry data illustrating the gating strategy and definition of subsets analyzed should be included.

This has been clarified. Fluorochromes, antibody panels, and cytometer configuration have been noted. In addition, two new supplementary data have been provided. One table describing the fluorochromes for NK and DC (Supplementary Table 1) and one Figure showing the gating strategy (Supplementary Figure 5).

Monocytes are described as CD3+ in the text and figures, which is not appropriate.

That was an unintentional typo that has been corrected in the text and in the Figure legend (Supplementary Figure 4).

Activation marker expression should be presented as a percentage of each cell population rather than as a percentage of total PBMC, and the distribution of cell subsets within PBMC should be included.

We apologize for our mistake in labeling the Y axis of Figure 6. The percentage of cells

Answers to the reviewers' comments

expressing CD69 refer to each cell population and not to the total PBMCs, and the distribution of subsets within PBMC is now presented in Supplementary Figure 5.

Figure 4e appears to indicate that NK cells frequently constituted >50% of PBMC (in some samples, >80%); this is unexpected and should be addressed.

Reviewer has made reference to the Extended Data Figure 4e. This was an oversight. Results should have been expressed considering NK cells gating as percentage of CD3, CD14, CD20 negative subset cells. This has been corrected and data revised for Supplementary Fig 4e and 4f as well.

Day 7 results of CD69 staining appear to be an outlier, and raise concern about technical problems with these samples; the authors should address quality control used to ensure that these results are valid.

It is intriguing that CD69 was reduced in all lymphocyte subsets on Day 7. However, data presented in Figure 3d, e, and f, which was produced by a different method, also show a reduction in white blood cells (particularly lymphocytes and neutrophils) occurring on day 7.

In addition, a similar pattern of reduction in the CD69 expression by Day 7 after ZIKV infection has been recently reported for central memory CD8⁺ and CD4⁺ cells and NK CD16⁺ and CD16⁻ CD56⁻. Please see supplementary Fig 5c,d and e in Osuna, C.E., et al. Zika viral dynamics and shedding in rhesus and cynomolgus macaques. Nature medicine (2016). We believe that the similarity between these results from two different groups confirm the pattern we are seeing. Taking into account of those profiles and our findings, it seems to be that reduction of CD69 by day 7 is common after ZIKV infection. In fact, we had our results before Osuna et. al published their results.

The timing of B and T cell activation observed is unusual and surprising, with significant changes at days 1 and 2 post-infection;

We agree with the reviewer that the pattern of early activation of T and B cells might look unusual. However, this pattern is also similar to the very early systemic increase of proinflammatory cytokines reported in Figure 8. This rapid onset of inflammation seems to be a characteristic of ZIKV infection.

The authors' statement that the "pattern of activation ... were similar to those previously reported" is not justified from the article cited.

This statement should be applied to the NK CD16⁺CD69⁺ cells. This was corrected in the current version and now reads as follows:

"Nevertheless, the pattern of activation of NK CD16⁺CD69⁺ we observed in our macaques is very similar to the pattern previously reported for rhesus macaques of Indian origin²⁵"

Please see supplementary Fig 5e in Osuna, C.E., et al. Zika viral dynamics and shedding in rhesus and cynomolgus macaques. Nature medicine (2016).

6. In several places in the manuscript, the authors refer to virus-specific "total IgG". It is not clear

Answers to the reviewers' comments

how this was defined; ELISA OD values are only indirect measures of total IgG and this distinction should be kept clear throughout.

We agree on this observation. We found only one instance referring to total IgG on page 18 in the original version.

“This is in agreement with the lack of induction of DENV IgM and the moderate expansion of the ZIKV IgG titers (total or to NS1 protein)....”

This has been changed to:

“This is in agreement with the lack of induction of DENV IgM and the moderate expansion of the ZIKV IgG titers (against the whole virus or NS1 protein)”

The discussion of “fold increases” in OD values is not a good measure of antibody responses, in particular for comparing flavivirus-naïve and flavivirus-immune groups, and should be deleted.

We agree with the reviewer partially on this observation. While the “fold increases” in OD values is not a good measure of antibody responses, the pattern of OD value expansion (fold increase) is quite different during a primary and a secondary DENV infection. Indeed, there are very well characterized profiles to differentiate from a primary to a secondary DENV infection supported in the explanation of the IgG levels measure in OD.

The graphics below show the classical pattern representing the evolution of the antibody response in a primary versus a secondary dengue infection (source CDC website)

Legend

NS1	DENV-reactive IgG
Virus	DENV-reactive IgM
IgM	Dengue viral protein, NS1
IgG	

The fact that the IgG to dengue after Zika infection (measured in OD) in naïve animals is ten-fold higher than the OD values observed in dengue pre-exposed animals is a very important observation. This finding confirms that ZIKV infection is able to induce high level of cross-reacting antibodies to dengue virus (being a cofounding factor affecting the diagnostics of ZIKV). On the other hand, the limited increase in the dengue IgG levels (measured as OD) after ZIKV infection in animals previously exposed to dengue confirms a very important fact of this work: Zika virus does not behave as a secondary dengue infection after a primary exposure to DENV. We have modified the way we stated the results but we do thing that it is still important to highlight this fact.

Previous statements read:

Thirty days after infection all ZIKV-exposed macaques showed a significant increase in the cross-reacting anti-DENV IgG values. However, the expansion of the OD 450nm values related to baseline was 10-fold increased in naïve macaques compared to DENV-immune macaques (11.9

Answers to the reviewers' comments

vs. 1.8 fold increase respectively) (Extended Data Figure 3c).

New text reads:

“Thirty days after ZIKV infection, all exposed macaques showed a significant increase in the cross-reacting anti-DENV IgG value. Interestingly, naïve animals had ten times in OD 450nm values compared to their own IgG baseline levels (Supplementary Fig. 3c)”.

We agree with the reviewer that the statement “fold increase” has less value when evaluating the IgG values for ZIKV NS1 protein.

This has been changed to:

“The titers for IgG anti-ZIKV-NS1 protein were also enhanced in both cohorts (Supplementary Fig. 3e).

7. Several hematologic and immunologic parameters showed significant differences between DENV-immune and DENV-naïve animals at baseline (day 0), e.g., neutrophils, IFN-alpha, CXCL13. This should be addressed as a potential confounding factor.

Yes, this is a very important observation. It is not clear to us why some immunologic and hematologic parameters are consistently high at baseline in the naïve or in the pre-exposed cohorts. All animals were handled in the same way during the entire ZIKV-infection studies and in the initial forty days of quarantine. Naïve animals were selected in the same range of age as the pre-exposed animals (Now described in the text). For examples, all four naïve animals had statistically high value of activated B cells at baseline compared to the four pre-exposed animals. Furthermore, WBC count and absolute lymphocyte and neutrophil numbers were significantly higher in the pre-exposed cohort. For this reason, in some instances we do compare the changes on different immunological or hematological parameters with their own basal levels more than with the values from the control (naïve) cohort.

Minor comments:

8. Abstract- The conclusion that CXCL13 might be useful as a biomarker of acute ZIKV infection is not warranted.

We agree with the reviewer that this conclusion is not warranted. However, the results of another work, also in rhesus macaques where the authors used a mix of proteins and adjuvants, clearly showing that CXCL13 could be used as a biomarker for germinal center activity called our attention. Our findings of high levels of CXCL13 in the serum of both cohorts after ZIKV infection was unexpected and remarkable. We wanted to highlight these observations. Although, this conclusion has been removed from the Abstract, it is still mentioned in the Discussion section.

The reference to “in response to stimuli” for CD8/CD107a cells should be clarified. The Abstract should end with a conclusion.

We clarified the reference:

Now it reads: (to ZIKV envelope, NS1 peptides, and the whole DENV)

Answers to the reviewers' comments

Abstract has been modified accordingly.

9. Author summary- The authors should clarify the meaning of “results in severe forms”.

Clarified in the text. This has been changed to:

“...results in severe clinical presentations including hemorrhages and shock”.

10. Introduction- The concluding paragraph should be revised to more specifically address the objectives of the present study.

We have revised the whole last paragraph of the Introduction accordingly

11. Page 6 line 3- “quarantine period” is not explained in the Methods section.

This has been clarified and now reads:

Prior to Zika infection all eight animals were subjected to a forty-day quarantine period.

12. Page 6 line 22- Re: “significant” decrease in neutrophils, it appears that baseline neutrophil counts were higher in this group; this should be noted in the text as a possible explanation for the difference.

We have acknowledged this above under Major findings and have noted in the text as a possible explanation for the difference.

The new text reads as follows.

Interestingly, the basal level of those cells were also significantly higher in cohort 1 animals than in cohort 2”

13. Page 6 line 27- Supplemental figure 4f does not show an increase in CD14⁺CD16⁺ cells as described in the text.

We have revised this accordingly. The increase was observed only in the subset CD20⁺CD3⁺CD14⁺

14. Page 9 line 4- The meaning of this sentence (“titers for anti ZIKV-NS1 protein were also expanded”) is unclear.

We rewrote this sentence. Now it reads:

However, macaques pre-exposed to DENV-2 (n=4), showed an increase of ZIKV-NS1 IgG at days 30 and 60 after DENV infection. Three out of four macaques in group 2 (DENV-2-pre-exposed) still had detectable cross-reactive IgG to ZIKV-NS1 protein at 1 and 2.5 years p.i. (Supplementary Fig. 3d).

15. Page 9 lines 29, 32- The use of “endpoint” is incorrect in these sentences (“higher endpoint dilution titers in the first two dilutions”).

Reviewer's observation is correct.

Answers to the reviewers' comments

Previous descriptions was:

“For samples collected 30 d.p.i, naïve macaques had ZIKV-specific IgG endpoint dilutions titers up to $1 \times 10 \log_3$ and the DENV-immune cohort showed significantly higher endpoint dilution titers in the first two dilutions ($p=0.0286$ and $p=0.057$) when compared to naïve macaques (Fig 4b). DENV pre-exposed macaques also experimented a statistically significant increase in endpoint dilutions titers compared to their own baseline values in the first two dilutions ($p=0.0293$ and $p=0.0064$). One of four macaques (MA071, DENV-2-exposed) showed the highest endpoint dilution ($1 \times 10 \log_4$)...”

We revised this description and now it reads:

“For samples collected 30 d.p.i, naïve macaques had ZIKV-specific IgG up to a dilution of $1 \times 10 \log_3$ and the DENV-immune cohort showed significantly higher titers in the first two dilutions ($p=0.0286$ and $p=0.057$) when compared to naïve macaques (Fig. 4b). DENV pre-exposed macaques also showed a statistically significant increase in the ZIKV-specific IgG titers compared to their own baseline values in the first two dilutions ($p=0.0293$ and $p=0.0064$). One of four macaques (MA071, DENV-2-exposed) showed the highest titer (up to a dilution $1 \times 10 \log_4$)”

16. Page 10 line 11- The meaning of “highest enhancement potential” is unclear.

The word “potential” was deleted and revised as “highest enhancement”. Now it reads “One DENV-2 pre-exposed animal (MA071, see below) showed the highest enhancement (>20%)”.

17. Page 10- Given the small sample sizes, it would be preferable to express frequencies (of animals with positive PCR assays) as fractions (e.g., “2 of 4”) rather than percentages; the latter are misleading.

We agree with the reviewer and we changed from percentages to number of animals.

18. Page 10-11- The low yield of PCR testing on urine is unexpected and worthy of comment by the authors.

We do not have a clear answer to this observation. However, the titers in urine are very much similar to two other groups published on ZIKV RNA in urine. The number of animals used in those studies are limited. Because of that, and with high respect to the reviewer’s observation, we do not believe that there is enough published data to compare our results objectively.

19. Page 11 line 21-24 and page 14 lines 23-25- It is not clear what data justify the conclusion that there was greater variability in results in the DENV-immune group than in the naïve group. The Discussion of this point (page 19) should be modified.

From our results there is a trend for such parameters to be grouped in the naïve animals while they are more dispersed in the pre-exposed animals. However, we agree that judging this variability is speculative.

Answers to the reviewers' comments

Text from page 11 has been deleted.

Text from page 14 has been deleted

Text from page 19 has been deleted

20. Page 12-13- The range of cytokine/CD107a frequencies in unstimulated background control samples should be noted.

We appreciate the reviewers comment. As the values for unstimulated control samples for each cytokine and CD107a varies per animal and per staining parameter we have included the information in the Supplementary Table 2.

21. Page 14 line 3- The data in figure 7c and 7f do not appear to justify the statement that CD8+ T cell CD107a responses were diminished on day 60 relative to day 30.

Thank you for pointing this out. We agree with the reviewer's observation and have modified the text accordingly.

22. Page 17 lines 5-6- This statement ("occurs only in about 1% of human population") is inaccurate; it refers to severe dengue illness rather than ADE.

This statement was deleted while shortening the discussion. We believe that this version reads much better.

23. Page 18- Can the authors suggest alternate hypotheses for the finding of anti-ZIKV NS1 in DENV-2-immune animals but not DENV-1-immune animals? For example, could differences in the content of NS1 protein in the DENV inocula explain this result?

This is a great suggestion on our rationalization of the finding of anti-ZIKV NS1 in DENV-2-immune animals but not DENV-1-immune animals. We extensively reviewed the literature to provide a reasonable explanation. However, from the data available we have not been able to find any information (at genetic or structural level) that could support a plausible hypothesis. However, to answer the reviewer's comment, those animals were challenged 2.5 years ago (actually more than 3 years by now) with the same amount of virus for all serotypes 1×10^5 pfu. Particularly interesting is that the two animals exposed to DENV1 had significant low viremia and viremia days when compared to DENV 2 exposed animals. See the table below from our unpublished results. At this point, we have no other alternative than to consider this fact as logical hypothesis and we have already explained this in the new version.

In this revised version the text reads:

" One another plausible explanation is that the animals previously exposed to DENV2, 2.5 years before ZIKV infection, in spite of being challenged with the same amount of inoculum, had significantly more viremia days when compared to the group exposed to DENV1 (White, L.J. and Sariol, C.A. unpublished results).

Answers to the reviewers' comments

[UNPUBLISHED DATA REDACTED FROM PEER REVIEW FILE BY EDITORIAL TEAM UPON AUTHOR REQUEST]

24. Page 19-20- The authors have not presented data to claim a “perfect correlation” between BAFF and IFN- α production (page 19 line 16) or a “positive correlation” between CD20+/CD69+ B cells and serum levels of CXCL13 (page 20 line 5).

We have two separate observations. One,

“...IFN- α was found to be significantly higher, compared to basal levels only in naïve macaques at 24, 48 and 72 hrs post- infection (h.p.i.) ($p=0.004$, $p=0.02$ and $p=0.037$, respectively)”.

The second,

“B Cell-Activating Factor (BAFF) was massively produced in all but one animal (6N1) in the naïve cohort from 24 h.p.i. through 30 d.p.i. However, because the lack of BAFF secretion by that animal the values did not reach statistically significant difference compared to baseline levels”

We agree with the reviewer that this may not be a perfect correlation. However, the sustained statistical higher levels of IFN- α and BAFF in most of the naïve animals, when compared to pre-immune animals, are of high interest due to the role of IFN- α in the induction of BAFF. We humbly request to reconsider that this link should not be avoided.

In this version we changed this statement to:

“The massive early production of BAFF in the naïve cohort after ZIKV infection is remarkable when compared to the high statistically significant production of IFN- α in those macaques in contrast to the DENV-immune macaques.”

Answers to the reviewers' comments

25. Page 20 lines 29-30- It is not clear what the authors intend by this statement (“This confirms the relevance...”). They have not presented evidence that lower viremia in DENV-immune animals (which was not statistically significant) is related to cell-mediated immunity.

The reviewer's observation is very useful to change the way we present these results.

We found that CD8⁺ cells in dengue-immune animals trend to produce higher levels of CD107a at Day 30 and 60 after being stimulated with ZIKV and dengue virus antigens. The difference is stronger at 60 days p.i. At the same time, we found that the production of perforin, produced by CD8⁺ CD107a⁺ cells reached statistically significant difference by 48 h.p.i ($p=0.033$), and maintained until 60 d.p.i. ($p=0.045$) compared to basal levels only in animals previously exposed to dengue. It is difficult to ignore this functional relationship. However, we cannot link those findings with the trend of having lower viremia in the pre-immune animals. We changed the statement and now reads as follows:

“These findings suggest an essential role of the cell-mediated immune response to ZIKV infection in subjects previously exposed to DENV”.

26. Page 21- There is limited scientific value to a single instance of rash; the Discussion of this point should be significantly curtailed.

We agree with the reviewer. We were unable to confirm or rule out Zika as the cause of the nonspecific rash in this animal. In this version we significantly shortened the discussion on this issue. Also we added this statement as per the reviewer's comment:

“However, we consider that there is limited scientific value to a single incidence of rash”.

27. Page 21 lines 22-23- The authors have not presented any data on “NK activity specific for ZIKV”.

The statement related to this particular point has been deleted.

28. Page 21 (“we are establishing that the presence of previous immunity to DENV shortens infection”)- Since results were not statistically significant, the authors should be more cautious in their conclusion.

We agree we should be more cautious when expressing such conclusion. However, we are sure the reviewer also agrees with us, from our results it is clear that previous immunity to dengue does not contribute to enhancing the course of ZIKV disease. There is no higher ZIKV viremia. No significant change in the hematology, clinical status, immune cells, or cytokine profiles that can suggest a deleterious effect of a previous immunity to dengue. Our results also indicate that ZIKV does not behave like a secondary dengue infection when it occurs in the period of time associated with the worst or severe secondary infections. On the contrary, we show that several parameters have a trend indicating that at least, after 2 years, previous dengue exposure may modulate the immune response in a positive way in controlling the course of ZIKV infection.

On this version we changed the statement as follows:

Answers to the reviewers' comments

“...our results provide valuable clues that the pre-existence of a mature immune response to DENV neither enhances the peak of ZIKV viremia nor impairs the course of its infection”.

29. Animal cohorts (page 26)- It is particularly important that the authors describe the animal cohorts more clearly. The authors should be more specific regarding matching for age. Figure 1 is not particularly helpful in showing the timeline of events and should be revised.

We have revised this Figure and provided more specific information.

30. Page 26 line 26- this presumably should read “seronegative for DENV and ZIKV”.

Modified

31. qRT-PCR (page 27)- The limit of quantitation should be defined for the qRT-PCR assay used. Further detail should be provided to justify inclusion of data on “highly suspicious positive” assay results, particularly evidence that these are not false positive results.

The level of detection has been described and the results that were highly suspicious have been deleted. However, we are very confident that the highly suspicious results were not false positives. The increase in the amplification curves, after the CT used as limit of detection, was very clear in the highly suspicious samples but not in the negative controls. We have deleted all the results related to saliva but kept only the “highly suspicious” positive urine results with a detailed justification that these are not false positive.

32. PRNT/FRNT assay (page 28)- The authors should clarify the use of mean focus diameter, whether the reduction measured was in focus/plaque diameter or number, and whether titers reported are endpoint titers or calculated using other methods. (Based on the values reported, it is likely that these are endpoint titers.)

This was an oversight and has been corrected. Clearly, the reduction in neutralization was expressed as reduction in the focus/plaque numbers and not diameter. And yes, this was calculated as endpoint titers.

33. Immune response assays (page 30)- The peptides used should be described in additional detail.

Peptides have been described.

34. Figure 2b- The y axis scale should be revised, e.g., 90 to 105 degrees F. It would be considered more appropriate scientifically to express temperature in degrees C. Also, rectal temperatures (supplemental figure 1a) would be preferable to include in the main results in place of figure 2b.

Changes have been made in the revised version as per the reviewer's advice. Rectal temperature was included in the main results in place of Figure 2b.

Answers to the reviewers' comments

35. Figure 3 duplicates data from Table 1b; it is unnecessary to present these data in both table and graph formats. Figure 3c does not add to the manuscript and can be eliminated.

We have eliminated Figure 3 and presented the data in Table 1b, now named Table 1

36. Figure 4- The y axes in panels a and b should be labeled "OD".

Corrected

37. Figure 4c- The x axis should not use both logarithmic notation and a label stating "log₁₀ reciprocal value").

Figure 4c is now Figure 1. The notation "log₁₀ reciprocal value" has been removed.

38. Figure 5a duplicates data presented in Table 1a; it is unnecessary to present these data in both table and graph formats.

We agree with the reviewer that the data presented in both Figure 5a and Table 1a (now Table 2) are the same. However, Figure 5 summarizes all the data in Table 2 visually and shows a general trend in a concise way. These data are crucial to our manuscript and we strongly believe that this will help the readers to see it both ways.

39. Figure 5b- The y axis is confusing; the major tick labels are linear but logarithmic minor tick marks are shown. This figure should adopt a standard logarithmic scale without breaks as in Figure 5a.

It has been very difficult for us to generate a proper Y axis for this Figure. Due to the number of time points reported as zero (0), the program we are using (Prism) did not allow to represent all the time points in standard logarithmic scale as in Figure 5a.

We have decided to present this data in a Table. In this version it is Table 3.

40. Figure 7- The corresponding panels for day 30 and day 60 responses should utilize the same y axis range to facilitate comparisons.

Thank you for pointing this out. We agree with the reviewer's observation and have modified Figure 7 accordingly.

41. Table 1- Tables 1a and 1b present different information and should be separated.

In this version, they are separated as two Tables.

42. Supplemental figure 2- The micrographs shown need better description of the specimen and the staining method.

Answers to the reviewers' comments

Figure 2c has been removed. Current Figure 2c has been reoriented to show the epidermis on the top, and the description of the Figure improved as follows:

“(c) Hematoxylin and eosin light micrograph of the haired skin of this macaque examined at 40x magnification. Few superficial dermal capillaries exhibit low numbers of perivascular mononuclear inflammatory infiltrate with a predominant population of lymphocytes but lesser plasma cells and histiocytes. Endothelial lining of capillaries is slightly hypertrophied and few dermal capillaries are surrounded by increased space with subtle separation of dermal collagen bundles adjacent to the vessels (edema).”

43. Supplemental Figure 3- Since OD values are already presented in panels c and e, the information in panel f is unnecessary.

Panel f has been removed.

Reviewer #3 (Remarks to the Author):

Pantoja-Maldonado et al. reported on pre-existing dengue immunity does not affect Zika pathogenesis but modulates the immune response in rhesus macaques. This subject is an important issue to the field of flaviviruses for many decades, especially for dengue virus (DENV) and Zika viruses (ZIKV) infections. The current well-designed investigations and in depth analysis of the insignificant effects of pre-existing dengue immunity on ZIKV infection provide an essential message on the argument of the role of in vitro artifact of antibody dependent enhancement (ADE) in disease pathogenesis induced by flaviviruses. The authors singled out critical information on the small number of rhesus macaques utilized in early days to investigate the ADE, coupling mainly with in vitro experimental studies, the hypothesis since then has become a legend, but constantly being debated and challenged. The current in vivo data will surely likely put out the hypothesis mainly derived from in vitro in peace to the field of DENV and ZIKV infections. One of the worth mentioning is that the observed profile on the faster clearance off ZIKV in pre-existing dengue immunity was similar to that of seen in dengue patients in secondary infection, in which the DENV was cleared off faster in secondary infection 1. As such, the in vivo information provided in the current manuscript will make a giant step forward to the pathogenic causes derived from DENV and ZIKV infections, so do vaccine development and control.

We could not agree more with this reviewer on this critical answer to the crux of the argument of the role of in vitro artifact of antibody dependent enhancement (ADE). We also agree with the reviewer on the similarity between fast clearance off ZIKV in the presence of pre-existing dengue immunity and what is seen in dengue patients in secondary infection. In the revised version this has been mentioned accordingly and the suggested reference included.

Minor comments:

1-The authors mentioned during quarantine period; Did the quarantine come after the DENV infection or naive animals prior to DENV followed by ZIKV infections?

This has been modified in the revised version in order to present this information clearly. Under Animal's cohort heading we added the following statement:

Answers to the reviewers' comments

"Prior to Zika infection all eight animals were subjected to a forty-day quarantine period"

2. Measuring the body weights of rhesus monkeys required to put down the animals, will the procedure effect on the reading and the observed outcomes reported here?

This is an interesting comment and we have been asked about this before. It is difficult to answer this question, as control animals without anesthesia should also need to be handled similarly. However, it is very difficult to get IACUC approval for this type of control. The best argument we have is that both control and experimental animals (in this case dengue pre-exposed and naïve) were handled the same way and were administered similar amount of anesthesia, even though, we have always been able to find significant differences between or among groups.

3. The temperatures were measured with an infrared device, how accurate the readings were? Were the readings affected by distance and angle of the measurements?

We thank the reviewer for this observation. We added this comment under the heading Clinical status. Since the animals were already anesthetized, it allowed for accurate and constant positioning of the IR thermometer during each reading. As directed by the IR thermometer manufacturer, readings were always taken at close range (within 2 inches) by pointing the thermometer directly to the animals' forehead.

4. Despite the platelet counts remained constant, was the functionality of the platelets the same?

We acknowledge the relevance of this observation. However, because our results showed that the platelet values were not changed and no symptoms of bleeding or other platelet malfunctioning were observed, we did not perform platelet test to determine their functionality. We are running another similar studies with animals with shorter period of exposure to dengue. There, we will take this useful suggestion and implement a platelet function test.

5. With the notion that asymptomatic infection and latent of DENV infection may occur in rhesus monkeys, hence, hence the animal with rash may shed DENV. It may be helpful to sort out whether DENV vRNA was in the collected specimens that were tested samples for ZIKV vRNA?

We thank the reviewer for this very specific observation. While latent DENV infection cannot be rule out, this phenomenon is very rare and poorly characterized. We are aware of only a few published work dealing with this uncommon behavior of dengue virus. In this particular work animals were exposed to DENV 2.5 years before they were exposed to ZIKV. This period of time makes it very unlikely for survival of DENV in any tissue. However, the animal with rash (BS14) produced similar or lower IgG titers or neutralizing antibodies to DENV 30 days after ZIKV infection. In this version we have added data on the neutralizing titers 60 days p.i. The Neut titers against all four DENV fade out more rapidly in this animal than the other three DENV-exposed ones. These results do not support the notion of a recent dengue replication in this animal.

1. Vaughn, D.W., et al. Dengue in the early febrile phase: viremia and antibody responses. J Infect Dis 176, 322-330 (1997).

REVIEWERS' COMMENTS:

Reviewer #2 (Remarks to the Author):

The authors have addressed many of the points raised, but I have remaining concerns:

Previous comments #1&2 re: writing- The previous issues raised were numerous proofreading and grammatical errors, poorly communicated concepts, and lack of focus on key findings. The revised manuscript has made little headway on these issues, which remain too numerous to catalog but in many cases should be quite obvious to a casual reader. The Abstract has been improved, but the Results and Discussion sections are overly long and unfocused.

Previous comment #4 re: statistically significant results- Contrary to the authors' reply, I do not find the statistical tests to be clearly noted; a specific example is the section on "Cytokine profile", but this is not the only such example. Also, the authors should consider that use of 4 significant digits is not appropriate when reporting p values with such small sample sizes.

Previous comments #6&15 re: OD values vs. IgG titers/levels and reference to endpoint titers- The authors continue to conflate these measures inappropriately. For example, the discussion of "IgG values ... in the first two dilutions" (page 11) is confusing. Serial dilutions obviously do not change the relative concentrations of an analyte between two samples; the author mean "OD values" rather than "IgG values". I take particular issue with the authors' new conclusion that ZIKV "does not behave as a secondary dengue infection". I believe the authors are referring to the observation that OD values in the DENV IgG ELISA increased by a mean of 1.8-fold. However, the neutralizing antibody data in Table 1 convincingly show that the antibody response to DENV did appear as a secondary dengue infection. An 11-fold increase in OD from a very low value (in DENV-naïve animals) cannot be easily equated to a 1.8-fold increase from a very high value (in DENV-immune animals).

Previous comment #24 re: correlation between BAFF and IFN-alpha- I do not see how the revised text improves the meaning. If the authors simply wish to point out that elevated IFN-alpha and BAFF levels may be linked based on published information, that might be a reasonable statement.

Previous comment #25 re: link between CD107a staining and lower viremia- I do not feel that the revised text addresses this concern. The reference to an "essential" role is not supported by data. The statement should be deleted or rephrased.

Reviewer #3 (Remarks to the Author):

The revised manuscript has partially addressed the concerns of this reviewer. For asymptomatic issue, as indicated in the rebuttal letter, that the authors stated that latent DENV infection cannot be ruled out, and in conjunction with statement that low Neut or no titers to DENV at 60 days after ZIKV infection as well as 8 animals shown within the rebuttal letter (BS14, BP71, BS17, BS81, BP30, BP37, BP38, and BP36) demonstrated low Neut titers at 10 days post challenge, the authors should discuss a possibility that these animals could be in the category of hypo-immune response. Hence the low antibody response could not rule out the occurrence of latent DENV in these animals and DENV replication in tissues of these animals.

Answers To Reviewers' Comments

Reviewer #2 (Remarks to the Author):

The authors have addressed many of the points raised, but I have remaining concerns:

Previous comments #1&2 re: writing- The previous issues raised were numerous proofreading and grammatical errors, poorly communicated concepts, and lack of focus on key findings. The revised manuscript has made little headway on these issues, which remain too numerous to catalog but in many cases should be quite obvious to a casual reader. The Abstract has been improved, but the Results and Discussion sections are overly long and unfocused.

The language and grammar were revised. The main concepts have been reviewed and rewritten to reinforce the value of our findings. Results and Discussion sections were thoroughly reviewed.

Previous comment #4 re: statistically significant results- Contrary to the authors' reply, I do not find the statistical tests to be clearly noted; a specific example is the section on "Cytokine profile", but this is not the only such example. Also, the authors should consider that use of 4 significant digits is not appropriate when reporting p values with such small sample sizes.

We are sorry we were not clear enough answering this comment. Statistical tests were described under Statistical Method section. Now we further described the tests providing more details. In addition, the specific statistical tests are mentioned in the figure legends where statistical differences were established. We reviewed the use of 4 significant digits and now the p values are expressed in relational terms with the alpha values and p values less than 0.01 are expressed as $p < 0.01$. As before the significance threshold for all analyses was set at 0.05.

Previous comments #6&15 re: OD values vs. IgG titers/levels and reference to endpoint titers- The authors continue to conflate these measures inappropriately. For example, the discussion of "IgG values ... in the first two dilutions" (page 11) is confusing. Serial dilutions obviously do not change the relative concentrations of an analyte between two samples; the author mean "OD values" rather than "IgG values".

In principle we do agree with the reviewer's comment on that the effect of serial dilutions do not change the relative concentration of an analyte. However in our binding assay (or any other similar) the antibody-antigen complex formation can be controlled by varying the concentration of antibody relative to the concentration of antigen. By using a dilution series it is possible to determine the relative end point titers of ZIKA specific or cross-reacting antibodies in our samples.

We rephrased this statement for clarity and consistency with reviewer's observation.

The current statement reads:

"DENV pre-exposed macaques also showed a statistically significant increase in the OD values while measuring IgG endpoint titers for binding to ZIKV when compared to their own OD baseline values in the first two dilutions ($p < 0.02$ and $p < 0.01$). One of four macaques (MA071, DENV-2-exposed) showed the highest OD for serum dilution up to 1×10^5 ".

I take _particular_ issue with the authors' new conclusion that ZIKV "does not behave as a secondary dengue infection". I believe the authors are referring to the observation that OD

values in the DENV IgG ELISA increased by a mean of 1.8-fold. However, the neutralizing antibody data in Table 1 convincingly show that the antibody response to DENV *_did_* appear as a secondary dengue infection. An 11-fold increase in OD from a very low value (in DENV-naïve animals) cannot be easily equated to a 1.8-fold increase from a very high value (in DENV-immune animals).

Reviewer's observation is correct. In animals with previous DENV exposition, the antibody response to DENV did appear as a secondary dengue infection after ZIKV exposition. Previous comment that was not clear to the reviewer, it was actually related to the Neut titers to ZIKV and not to the Neut titers to DENV.

In this revision we are highlighting this result and now it reads as follow:

"In subjects with previous immunity to DENV, we are showing that after 30 days, ZIKV infection can induce a transient 4-fold expansion of the Nabs titers against DENV reminding a secondary DENV infection"

Previous comment #24 re: correlation between BAFF and IFN-alpha- I do not see how the revised text improves the meaning. If the authors simply wish to point out that elevated IFN-alpha and BAFF levels may be linked based on published information, that might be a reasonable statement.

We are summarizing this finding in a more objective/descriptive way:

"Considering published data on the role of IFN- α in the induction of BAFF^{1,2} the early production of BAFF observed in the naïve cohort after ZIKV infection is noteworthy when compared to the high statistically significant production of IFN- α "

Previous comment #25 re: link between CD107a staining and lower viremia- I do not feel that the revised text addresses this concern. The reference to an "essential" role is not supported by data. The statement should be deleted or rephrased.

The statements related to CD107 has now been rephrased as follow:

"These findings suggest an effect of previous exposition to DENV in the cell-mediated immune response to ZIKV infection"

Reviewer #3 (Remarks to the Author):

The revised manuscript has partially addressed the concerns of this reviewer. For asymptomatic issue, as indicated in the rebuttal letter, that the authors stated that latent DENV infection cannot be ruled out, and in conjunction with statement that low Neut or no titers to DENV at 60 days after ZIKV infection as well as 8 animals shown within the rebuttal letter (BS14, BP71, BS17, BS81, BP30, BP37, BP38, and BP36) demonstrated low Neut titers at 10 days post challenge, the authors should discuss a possibility that these animals could be in the category of hypo-immune response. Hence the low antibody response could not rule out the occurrence of latent DENV in these animals and DENV replication in tissues of these animals.

To rule out a reactivation of a latent dengue infection, serum samples collected in the first ten days after ZIKV infection were tested by RT-PCR for the presence of DENV1 and 2 (The challenge serotypes for Cohort 1 animals).

All samples were negative and reported in the supplementary Table 9. In the absence of DENV replication is very unlikely that the rash presented by subject BS14 should be induced by a potential reactivation of a local latent skin-DENV infection. Related to the hypo-immune status of animals previously exposed to dengue we humbly disagree. As showed in the table provided in previous answers to the reviewers the DENV neutralizing titers, only 10 days after dengue infection, were robust and in the range needed to induce protection in macaques exposed to dengue in the course of different experimental protocols (Sariol and White, 2013)³. The DENV Neut₅₀ titers, after 10 days of DENV challenge, for the animals included in this work were:
BS14: 1:347, BP71: 1:275, MA071: 1:3,391 and BS02: 1:1,007
Titers were very similar for macaques challenged with DENV1 and 3 and for those exposed to DENV2 and 4.
All those macaques and results are part of an unpublished manuscript we are still working on. Partial data, relevant to this work have been provided as supplementary Table 3.

- 1 Gomez, A. M., Ouellet, M. & Tremblay, M. J. HIV-1-triggered release of type I IFN by plasmacytoid dendritic cells induces BAFF production in monocytes. *Journal of immunology* (Baltimore, Md. : 1950) **194**, 2300-2308, doi:10.4049/jimmunol.1402147 (2015).
- 2 Ittah, M. et al. Induction of B cell-activating factor by viral infection is a general phenomenon, but the types of viruses and mechanisms depend on cell type. *Journal of innate immunity* **3**, 200-207, doi:10.1159/000321194 (2011).
- 3 Sariol, C. A. & White, L. J. Utility, limitations, and future of non-human primates for dengue research and vaccine development. *Frontiers in immunology* **5**, 452, doi:10.3389/fimmu.2014.00452 (2014).